# Inadequate sanitation in healthcare facilities: A comprehensive evaluation of toilets in major hospitals in Dhaka, Bangladesh

**Nuhu Amin**[1,2]*, **Tim Foster**[1], **Md. Imam Hossain**[2], **Md Rezaul Hasan**[2],
**Supriya Sarkar**[3], **Aninda Rahman**[4], **Shaikh Daud Adnan**[3], **Mahbubur Rahman**[2],
**Juliet Willetts**[1]

**1** Institute for Sustainable Futures, University of Technology Sydney, Ultimo, NSW, Australia, **2** Health System and Population Studies Division, Environmental Health and WASH, International Centre for Diarrhoeal Disease Research, Bangladesh (icddr,b), Dhaka, Bangladesh, **3** Hospital Services Management, Directorate General of Health Services (DGHS), MoH&FW, Mohakhali, Dhaka, Bangladesh, **4** Communicable Disease Control (CDC) Program, Directorate General of Health Services (DGHS), MoH&FW, Mohakhali, Dhaka, Bangladesh

* mdnuhu.amin@student.uts.edu

**Data Availability Statement:** All relevant data are within the paper and its Supporting Information files.

## Abstract

### Background

Lack of access to functional and hygienic toilets in healthcare facilities (HCFs) is a significant public health issue in low- and middle-income countries (LMICs), leading to the transmission of infectious diseases. Globally, there is a lack of studies characterising toilet conditions and estimating user-to-toilet ratios in large urban hospitals in LMICs. We conducted a cross-sectional study in 10-government and two-private hospitals to explore the availability, functionality, cleanliness, and user-to-toilet ratio in Dhaka, Bangladesh.

### Methods

From Aug-Dec 2022, we undertook infrastructure assessments of toilets in selected hospitals. We observed all toilets and recorded attributes of intended users, including sex, disability status, patient status (in-patient/out-patient/caregiver) and/or staff (doctor/nurse/cleaner/mixed-gender/shared). Toilet functionality was defined according to criteria used by the WHO/UNICEF Joint-Monitoring Programme in HCFs. Toilet cleanliness was assessed, considering visible feces on any surface, strong fecal odor, presence of flies, sputum, insects, and rodents, and solid waste.

### Results

Amongst 2875 toilets, 2459 (86%) were observed. Sixty-eight-percent of government hospital toilets and 92% of private hospital toilets were functional. Only 33% of toilets in government hospitals and 56% in private hospitals were clean. A high user-to-toilet ratio was observed in government hospitals' outpatients service (214:1) compared to inpatients service (17:1). User-to-toilet ratio was also high in private hospitals' outpatients service (94:1)

**Funding:** This project has been funded by the Bill & Melinda Gates foundation (BMGF) [grant number INV-045360]. The funders had no role in study design, data collection and analysis, decision to publish, or preparation of the manuscript.

**Competing interests:** The authors have declared that no competing interests exist.

compared to inpatients wards (19:1). Only 3% of toilets had bins for menstrual-pad disposal and <1% of toilets had facilities for disabled people.

## Conclusion

A high percentage of unclean toilets coupled with high user-to-toilet ratio hinders the achievement of SDG by 2030 and risks poor infection-control. Increasing the number of usable, clean toilets in proportion to users is crucial. The findings suggest an urgent call for attention to ensure basic sanitation facilities in Dhaka's HCFs. The policy makers should allocate resources for adequate toilets, maintenance staff, cleanliness, along with strong leadership of the hospital administrators.

## Introduction

Globally, approximately 10% of healthcare facilities (HCF) have no sanitation services, meaning that 780 million people seek healthcare at facilities without toilets [1]. This problem is even more significant in low- and middle-income countries (LMICs), where 47% of HCFs lack basic water services (i.e., an improved water source on premises) and only 21% of HCFs have basic sanitation services. Basic sanitation service is defined as an HCF with improved latrines or toilets that are usable, separated for patients and staff, segregated for women with menstrual hygiene facilities, and meet the needs of people with limited mobility [1]. Ensuring basic sanitation for all is a priority for countries to achieve sustainable Development Goals (SDGs) by 2030 [1]. Furthermore, the availability and effectiveness of such services in HCFs is critically important given the type of healthcare services provided by HCFs. The absence of adequate water, sanitation and hygiene (WASH) services significantly contributes to substandard healthcare, influencing infection rates, service utilization, staff productivity, and the dignity of patients and healthcare workers [2]. Several studies suggest that the enhancement of WASH in HCFs could potentially prevent millions of deaths and lead to substantial financial savings in the order of billions of dollars [2,3].

Access to a functional and clean toilet plays a vital role in achieving safe sanitation services in HCFs [4]. The JMP report highlighted that although many HCFs have toilets on their premises, a significant number of them are non-functional and unusable [5]. Recent estimates suggested that globally, only 69 countries had estimates for coverage of improved sanitation facilities (toilets designed to hygienically separate excreta from human contact) in HCFs [6]. However, there is a global lack of data pertaining to toilet functionality and cleanliness, especially for large cities in LMICs.

Toilet hygiene is important in the control of infectious diseases associated with enteric and airborne pathogens in both HCFs and community settings [7]. Several studies have reported disease outbreaks in HCFs where the toilet has been identified as a source of infection [8–10]. Numerous enteric pathogens are found in high concentration in stools and therefore in toilets after defecation, particularly during disease outbreaks [7]. Unhygienic and dirty toilets pose significant concerns from both a public health and Infection Prevention and Control (IPC) perspective. Unhygienic and dirty toilets serve as breeding grounds for pathogens, including bacteria, viruses, and parasites, which can spread through contact with contaminated surfaces [8,9,11,12]. This risk is high in HCFs, where an unclean hospital environment including toilets contribute to healthcare-associated infections, particularly affecting immunocompromised patients [13]. Beyond health risks, unclean toilets compromise satisfaction, dignity and well-

being of both patients and staff [2]. Maintaining proper IPC measures demands clean environments to prevent transmission; however, unclean toilets hinder the ability to maintain IPC standards [13,14].

Bangladesh stands as one of the world's most densely populated countries (>1119 people per square kilometer), with about 22% of people living below the poverty line [15]. Although medical facilities are mostly concentrated in urban areas, the high population density in mega cities like Dhaka (population density: >23,000 people/km$^2$) exacerbates the challenges of managing healthcare services [16]. High patient flow could impact toilet functionality in HCFs [17,18]. However, there is a notable research gap concerning user-to-toilet ratio, particularly in HCFs within LMICs. A recent study conducted in six government hospitals in Dhaka during the COVID-19 pandemic revealed that the overall user-to-toilet ratio was high, at 25 in general hospitals [17]. According to the Bangladesh WASH standard-2021guideline, hospitals should maintain a 6:1 bed-to-toilet ratio [19] while WaterAid technical guide for construction of institutional toilets recommends for inpatient services, each ward should have one toilet designated for women and one for men. For outpatient services, there should be one toilet for every 20–25 patients or carers, up to the first 100 individuals. Beyond that, an additional toilet should be provided for every additional 50 patients or carers. Additionally, there should be a provision of a 2:1 ratio of female to male toilets [20]. The WHO-UNICEF JMP standard also indicates basic sanitation in HCFs covers at least one toilet dedicated for staff, at least one sex-separated toilet with menstrual hygiene facilities, and at least one toilet accessible for people with limited mobility [21]. It is crucial to examine the user-to-toilet ratio in large hospitals to assess how many of them comply with basic sanitation facility standards.

The burden of inadequate sanitation facilities in HCFs disproportionately affects women [22]. Women constitute over 70% of the health workforce [23], and they are also the primary users of healthcare services, particularly in primary care settings. Regrettably, sanitation services for women are often insufficient or lacking [22]. Ensuring equitable access to fundamental sanitation services and practices constitutes a concrete step toward promoting gender equality [22] and upholding the universal human rights to water, sanitation, and health. Sanitation facilities such as gender-segregated and hygienic toilets with waste disposal bins, as well as adequate washing facilities, are crucial for menstruating and postpartum bleeding women. Furthermore, there is a lack of global data concerning these parameters in HCFs [1].

Individuals with limited mobility also face significant challenges due to inadequate sanitation facilities in healthcare settings [22]. Accessible and inclusive sanitation services are essential for promoting the dignity and well-being of people with disabilities. Unfortunately, these facilities are often overlooked, leading to disparities in healthcare access and outcomes [22]. Addressing the specific needs of individuals with disabilities requires a comprehensive approach that includes the provision of facilities such as accessible toilets and appropriate washing amenities [24]. Despite the importance of these considerations, there is a lack of comprehensive global data regarding the availability and adequacy of such facilities in healthcare settings [1], limiting efforts to highlight and to address the issues.

There is a notable lack of global data on toilet functionality and hygiene, especially in LMICs such as Bangladesh. In Bangladesh, recent sanitation evaluations have focused on sub-district level hospitals, neglecting larger HCFs in major cities. No study has estimated the user-to-toilet ratio for different user groups (i.e., staff, patients, and caregivers) in major HCFs in LMICs. Furthermore, as noted above, there is a lack of data on sanitation facilities for persons with limited mobility and menstrual hygiene management (MHM) facilities for women in HCFs at local level. To address these critical knowledge gaps in HCFs, we conducted a cross-sectional study to comprehensively explore the availability, accessibility, functionality, cleanliness, user-to-toilet ratios, and adequacy of sanitation facilities in 10 government and two

private hospitals in Dhaka, Bangladesh. We also assessed toilet availability for women, staff, and individuals with limited mobility in all selected hospitals in Dhaka city to determine the number of hospitals with basic sanitation facilities that met their needs. The study focused on addressing existing research gaps and providing insights to enhance sanitation services and infection prevention in a high-density urban context in Bangladesh, with implications for other similar LMIC settings.

## Materials and methods

### Study design

We conducted a cross-sectional study between June and December 2022 (post-monsoon to winter season) in 10 government hospitals and two private hospitals. The data collection process took longer than anticipated due to delays in obtaining directors' approvals, and unavailability of patient records in certain hospitals. A team of six trained fieldworkers, supervised by the lead investigator (NA), conducted the data collection. We visited the selected hospitals one after another and made multiple visits as required to complete the data collection. On average, it took 18 days to collect data from each hospital. During this period, we conducted structured observations to assess the functionality and cleanliness of the toilets and estimate the number of users for each toilet (i.e., user-to-toilet ratio) in the selected hospitals. A structured observational checklist was used to assess the toilet functionality and cleanliness of the selected hospitals [25].

### Enrolment of study hospitals

All study sites were selected within Dhaka city. Before proceeding with the selection of study sites, we prioritized stakeholder engagement by conducting a meeting involving key stakeholders such as the Directorate General of Health Services (DGHS), Dhaka Water Supply and Sewerage Authority (DWASA), policymakers, and national and international NGOs. We revised our hospital selection criteria and data collection tools by incorporating suggestions from the stakeholder meeting. For this study, we only enrolled major tertiary care hospitals in Dhaka city. We included at least one hospital from each category (general, specialized, or medical college and hospitals) and of different sizes (small, medium, or large based on the number of beds). Additionally, we considered varied geographic locations within the Dhaka South City Corporation (DSCC) and Dhaka North City Corporation (DNCC) areas. A total of ten government hospitals were selected. This included two general hospitals, two medical college hospitals, and six specialized hospitals. For comparison purposes, we purposively selected two private medical college hospitals (Fig 1) to establish a reference group alongside the government hospitals. We did not include any primary-care hospital, and community clinics in this study.

### Operational definitions used to describe different variables in HCFs

For this study, where appropriate, we adopted the WHO-UNICEF Joint Monitoring Programme (JMP) definitions and extended the JMP definitions to describe additional sanitation-related variables relevant for HCFs in Dhaka city (Table 1). In cases where JMP or WASH-FIT (water and sanitation for health facility improvement tool) did not provide clear definitions for certain variables (i.e., clean toilets, toilet blocks, toilet users in HCFs, etc.,), we utilized our own operational definitions:

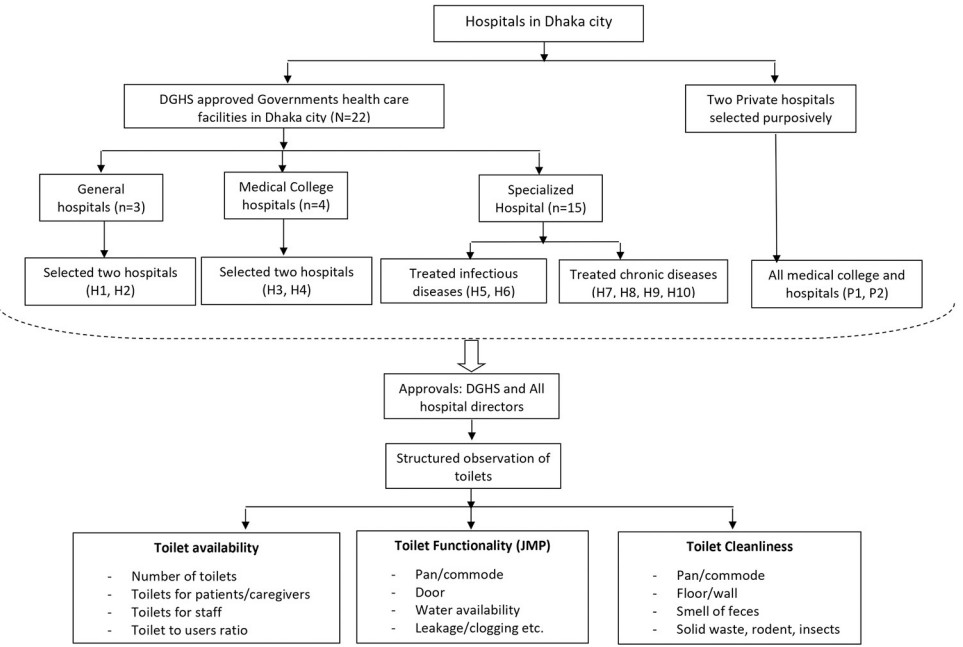

**Fig 1. Study flow chart.**

## Defining toilet blocks in each hospital floor

A "toilet block" was defined as a single toilet or group of toilets within HCFs that were used by a specific and defined group of people or users. After obtaining written approval from the hospital director, trained fieldworkers approached the head nurse of each floor/ward, who were responsible for overseeing the toilets and had a clear understanding of the number of toilets (i.e., availability), number of patients admitted, and caregivers present in their respective ward or duty area on the day of data collection. Additionally, we collected information on the total number of healthcare providers (i.e., doctors, nurses, cleaners) appointed to different shifts in the hospital. The objective of the study was explained to the nurses during this interaction. Following discussions with the nurse, one of the investigators (NA) listed total number of toilets present in each floor, and location of the toilets (i.e., inpatients ward or outpatient ward; patients or doctors, etc.).

The rationale behind dividing the toilets into multiple blocks on each floor was to ensure the collection of precise information on the number of users (patients, caregivers, and staff) from each toilet block. Toilet block was formed based on various factors such as the type of users (patients, staff, or both), patient services (inpatients, outpatients, and emergency), all patients wards (medicine, surgery, gynae, special care, etc.), and gender (male, female, or shared). For instance, if a toilet block was located near the surgery ward (a dedicated unit within a hospital where minor/major surgical procedures are performed, more than 10 patients stay within a room/space, require an overnight stay, or extended pre-and postoperative care), it was designated as the surgery ward toilet block. If there were two separate male and female toilet blocks for patients in one ward, they were considered as two distinct toilet blocks (referred to as block-1 and block-2). Each patient's cabin (one bed in an enclosed space/room with a private toilet) with an attached toilet was considered as separate toilet block and termed as the "toilet block for patients." For example, if there were 20 cabins on one floor, these toilet blocks were numbered as toilet blocks no# 1–20.

For healthcare professionals, such as directors or professors, they had access to private toilets inside their offices with maximum 1–2 users, termed as the "toilet block for staff." Similarly, several nurses and cleaners used one toilet and termed a "toilet block for staff." If there were five private toilets for doctors, one nurse toilet, and one toilet for other staff members, a total of seven toilet blocks were counted.

## Selection of toilets

The hospitals in Dhaka city exhibit a wide range of diversity, including variations in size (ranging from 200 beds to 2300 beds), building structures, and patient services, which were not consistent across the hospitals. The number and types of users also differed across hospitals.

**Table 1. Operational definitions used to describe different terminologies.**

| Terminology | JMP defined | Definitions used in this study |
|---|---|---|
| Basic Sanitation Service | Health care facilities with improved latrines or toilets which are usable, separated for patients and staff, separated for women with menstrual hygiene facilities, and meet the needs of people with limited mobility are classified as having a basic sanitation service. | Same as JMP definition |
| Usable toilet | The term usable here refers to toilets or latrines that are accessible to patients and staff (doors are unlocked or a key is available at all times), functional (the toilet is not broken, the toilet hole is not blocked, and water is available for flush/pour-flush toilets), and private (there are closable doors that lock from the inside and no large gaps in the structure). | The term "usable" refers to toilets or latrines that are accessible to patients and staff (doors are unlocked or a key is available at all times), functional (the toilet is not broken, the toilet hole is not blocked, and water is available for flush/pour-flush toilets), and private (there are closable doors that lock from the inside and no large gaps in the structure) and **toilet floor is not submerged with water or feces**[*] |
| Functional toilets | The toilet is not broken, the toilet hole is not blocked, and water is available for flush/pour-flush toilets | The toilet pan is intact and not damaged, the door is functional and equipped with a working lock, there is a consistent supply of water, no signs of leakage from the overhead sewage pipe were observed, the toilet pan was not obstructed, overflowed, or clogged, and toilet floor was not submerged with water. |
| Clean toilets | No JMP definition | Toilet cleanliness was assessed, considering visible feces on any surface of the toilet (including the commode/pan, walls, and floor), strong fecal odor, presence of sputum or saliva on any surface, the existence of flies, cockroaches, insects, and rodents, as well as the presence of solid waste and spider webs. The fieldworkers also evaluated toilet cleanliness based on their own perceived cleanliness scale during data collection, utilizing a six-point scale ranging from "not dirty at all" to "not usable" [26]. |
| Accessible toilet | Toilet doors are unlocked, or a key is available at all times | Toilet accessible during data collection (i.e., toilet is not locked, or key is available when requested/needed or toilet is not used as other purpose) |
| Gender segregated toilet | Toilet facilities that are specifically designated and labeled for use by males and females separately. | Gender-specific or gender-segregated toilet referred to separate toilet facilities designated for individuals of different genders. These facilities were designed to accommodate the specific needs and privacy requirements of both males and females. Gender-segregated toilets may not always be labelled as such, and hence were defined based on input from the head nurse or ward in charge. |
| Mixed-gender or Gender-neutral toilet: | A gender-neutral room with a single toilet is also considered as single-sex, as it allows women and men to use toilets separately [27]. | Mixed-gender use toilets, also known as gender-neutral, non-sex segregated toilet or unisex toilets, were facilities intended for use by individuals of any gender, accommodating both male and female users within a single toilet space. |
| Cleaners | "Staff responsible for cleaning" refers to non-health care providers such as cleaners or auxiliary staff, as well as health care providers who, in addition to their clinical and patient care duties, perform cleaning tasks as part of their role [27]. | Same as JMP definition |
| Operational definitions used in the manuscript | | |

*(Continued)*

**Table 1.** (Continued)

| Terminology | JMP defined | Definitions used in this study |
|---|---|---|
| **Toilet or toilet unit:** A single toilet, equipped with either a single raised commode for sitting or a single squatting plate, squatting pan, or sitting pan. | | |
| **Squatting pan flush toilet:** A squat toilet (or squatting toilet) is a toilet used by squatting, rather than sitting. This means that the posture for defecation and for female urination is to place one foot on each side of the toilet drain or hole and to squat over it. | | |
| **Sitting pan flush toilet:** A typical flush toilet is a ceramic bowl (pan) connected to a cistern (tank) that enables rapid filling with water, and also to a drainpipe that removes the wastewater. | | |
| **Toilet users in HCFs:** Toilet users were defined as any individuals (including healthcare professionals, cleaning staff, patients, their caregivers, and non-healthcare professionals) present at the hospital on the day of data collection, and who might reasonably be expected to desire to use the toilet during their time at the facility. | | |
| **Available toilets:** Total number of toilets present in the HCF, regardless of their functional status, whether they are open or locked. | | |
| **Shared toilets in HCFs:** Shared facilities refer to toilets within HCFs that are utilized by both healthcare professionals (including staff, clinicians, and administrative personnel) and patients. | | |
| **Inpatient ward/service:** An inpatient ward in a hospital is a designated area for individuals where patients admitted for treatment, stay overnight or for several days or weeks or even months due to more severe or complex medical conditions. It provides continuous monitoring, treatment, and nursing care in a hospital setting, distinct from outpatient services. In an inpatient ward, multiple patients typically share a room or designated area. | | |
| **Outpatients service/ward:** Outpatient services or wards refer to the areas or departments where treatment is provided to individuals who do not require overnight hospitalization. These services typically include medical consultations, diagnostic tests, and treatments on an ambulatory basis. Patients usually visit the facility and return home on the same day without being admitted for an extended stay. | | |
| **Cabin (Inpatients):** An inpatient private cabin in a hospital is a designated, enclosed space exclusively assigned to a single patient during their stay. It is designed to offer personalized care, privacy, and essential facilities, including a bed, seating, and medical monitoring and treatment resources. The cabin typically includes a private attached toilet with bathing facilities for both the patient and their caregivers. For the purposes of this study, each cabin is defined as a single "toilet block," considering that occupants exclusively use the toilet during their stay periods. | | |
| **Patients' caregiver or attendant:** In the context of Healthcare Facilities (HCFs) in Bangladesh, "patients' caregivers" refers to individuals who provide support, assistance, and care to patients receiving medical treatment in the healthcare facility. Caregivers may include family members, friends, or individuals designated by the patient to assist with activities of daily living, offer emotional support, and ensure the well-being of the patient during their stay in the healthcare facility. | | |
| **Hospital staff:** The term "staff" refers to individuals employed within the healthcare setting, including healthcare professionals such as doctors and nurses, administrative personnel, support staff, and/or cleaners. | | |

* The criterion "Toilet floor is not submerged with water or feces" was classified under the usable category. This decision was based on the recognition that the severity of this condition could categorize it as either an unclean toilet or a non-functional toilet. The presence of a submerged floor often resulted from a malfunctioning drainage system, overflow from the pan after flushing, or leakage from water pipes.

Considering the complexity and the potential for selection bias, the study undertook a census of all toilets in each hospital except for toilets located in the separate administrative buildings and toilets exclusively used by students or academic professionals. The benefit of using a toilet census in HCFs helped to provide a comprehensive and representative assessment of sanitation facilities, avoid selection bias, allowing for more accurate analyses and meaningful insights.

## Structured observation of the toilets

After finalizing the toilet blocks, fieldworkers utilized the Kobo ToolBox software (https://www.kobotoolbox.org/) to administer a structured observational checklist and collected information on the type of toilet block based on user (male, female or shared toilet) the number of toilets (availability) in each block, accessibility of the toilet (open or locked), type of toilet pan (squatting pan or sitting pan flush toilet), the functionality of the toilet pan or commode (broken, water seal present/absent, the functionality of the toilet door (broken or not broken), availability of water, overhead sewage pipe leaks, overflow of the pan, and toilet floor was not submerged with water [28]. Additionally, toilet cleanliness was assessed, considering visible feces on any surface of the toilet (including the commode/pan, walls, and floor), strong fecal odor, presence of sputum or saliva on any surface, the existence of flies, cockroaches, insects, and rodents, as well as the presence of solid waste and spider webs. The fieldworkers also evaluated toilet cleanliness based on their own perceived cleanliness scale during data collection, utilizing a six-point scale ranging from "not dirty at all" to "not usable" (S1 Fig in S1 File) [26]. Furthermore, the fieldworkers recorded information related to toilet facilities for persons with

limited mobility (i.e., wheelchair accessibility, wide door, metallic handrail, etc.) [29], and solid disposal bins inside the toilet for menstrual hygiene management (MHM). The definition of "toilets" for persons with limited mobility was adapted from Talib *et al* 2016 [29] and MHM facility of toilets were defined as presence of disposal bin inside the toilet [30]. We did not include the variable 'type of toilet' in our study because all the hospitals had toilet units designed as flush (cistern/bucket flush) toilets connected to septic tanks or ABRs through a piped system or directly discharged into the nearest drain.

## Estimation of toilet users

Toilet users were defined as any individuals (including healthcare professionals, cleaning staff, patients and their caregivers, and non-healthcare professionals) present at the hospital on the day of data collection, and who might reasonably be expected to desire to use the toilet during their time at the facility. To estimate the number of users in each toilet block, fieldworkers obtained patient's records from the head nurse (i.e., in-charge of a ward/floor) for that day. They also collected data to estimate the average number of caregiver(s) stay with the patient or visit the hospital each day.

To determine the gender-segregated toilets for both staff and patients, the fieldworkers initially observed if there was any signage (i.e., sticker) on the toilet block or door. In cases where no signage was present, they inquired with the duty nurse and cleaners regarding the predominant user group (i.e., male, female, or mixed-gender) for the toilet/toilet blocks. To calculate toilet users in the inpatient services, a 24-hour timeframe was considered, as healthcare professionals, patients, and their caregivers utilize the facilities throughout the day. For the outpatient department, operational hours were extended until 2:00 pm local time, and toilet users were accounted for during that duration each day. Data on toilet users were also gathered from specialized units, including laboratories, diagnostics, operation theaters, and intensive care units, using a similar methodology.

## Data analysis

Stata 15.0 (StataCorp LLC, College Station, Texas, USA) and Microsoft Excel 2019 were used for statistical analysis and graph preparation. To determine whether the data was normally distributed, Shapiro-Wilk, Shapiro-Francia, and Skewness and Kurtosis tests were used. To visualize the data, descriptive statistics were performed on the collected variables. Frequency and proportion were calculated for categorical variables such as toilet functionality (functional vs. non-functional), toilet cleanliness (clean vs. not clean), and user-to-toilet ratio. To calculate user-to-toilet ratio between inpatient and outpatient service, we considered the exact toilets user numbers on the day of data collection. We did not compare the user-to-toilet ratio between inpatient and outpatient services because outpatients may only be using the HCF for a few hours to a maximum of half a day, whereas inpatients would typically stay at the HCF for a full day or more. Mean and standard deviation were calculated to estimate the toilet functionality and cleanliness. Two sample Z test of proportions was used to determine whether the toilet functionality and cleanliness differ significantly between inpatient and outpatient services. The outcome and other relevant variables were analyzed and presented based on exposure variables. Descriptive statistics were also conducted separately for four population groups: doctors, nurses/other staff, patients/caregivers, and other toilet users. Univariate and multivariate logistic regression models were performed to evaluate the effects of multiple factors on the toilet functionality and cleanliness. Hosmer–Lemeshow goodness-of-fit test was used to assess how well the model is fit for the analysed data. Variance Inflation Factor (VIF) was performed to test the multicollinearity among the explanatory variables used in the model. The

association between toilet functionality, cleanliness, and hospital type, year of construction, type of users, inpatients or outpatients service/ward, gender, user number per toilet was investigated using the univariate analysis. In multivariate analysis, only the exposure variables were used that were found to be statistically significant in univariate analysis [31]. The multivariate model was adjusted for co-variates such as user type, gender, user number per toilet to overcome the confounding effects. The mixed-effects logistic regression model was also adjusted with robust standard error to control the clustering effect within hospitals. P-value < 0.05 was considered statistically significant.

### Ethical considerations

We obtained written informed consent from all participants in the study and received written approvals from the director general of health service (DGHS). We also obtained formal written approvals from all hospital directors to carry out the study. The study protocol received approval from both the ethical review committee of icddr,b, Dhaka, Bangladesh and the institutional review board of University of Technology Sydney, Australia.

## Results

### Description of the selected hospitals, number of beds and toilets

Among the 12 selected hospitals, there were two government general hospitals, two government medical college hospitals, six specialized hospitals, and two private hospitals. Among the selected hospitals, six were established before the year 2000. Overall, the number of beds ranged from 250 to 850, with an average of 422 beds per hospital. The highest number of observed toilets was in a 500-bed government medical college hospital (H4 = 430 toilets), while the lowest number of toilets was found in a 300-bed government specialized hospital (H5 = 43 toilets). Both private hospitals had 500 beds with 86 toilets in hospital P1 and 59 toilets in hospital P2. The total number of toilet users (staff, patients, caregivers, and others) varied across the hospitals, ranging from 826 to 20,899 users per day in each hospital, and on average, there were 2.2 caregivers per patients came to visit the hospital for patients care (Table 2).

### Description of toilet features, availability, accessibility, and water sources

Among 2875 toilets, 86% (n = 2459) were directly observed by the data collectors. The main reasons for not being able to observe (14%) toilets were either due to being locked (10%) or not obtaining permission to access the toilet. Among the toilets observed, we found 56% of toilets were equipped with a sitting flush toilet while the other 44% featured a low pan/squat designed toilets. Almost all toilets (99%) had a functional water seal. The flooring material in nearly all toilets was comprised of ceramic tiles (98%). Additionally, the water supply to all toilets was facilitated through a piped water system to cistern/tank flush (86%) or to faucet (14%) (Table 3). Overall, only 26% of the handwashing basins adjacent to patient toilets had water and soap available together for handwashing.

The flushing mechanisms in the toilets varied, with 68% utilizing cistern/tank flush systems, of which 18% were non-functional. In contrast, 14% of the toilets were pour/bucket flush. For anal cleansing, more than 51% of the facilities relied on both tap water and buckets, while 47% were equipped with a piped hand shower. Only 2% of the toilets provided tap water without a bucket. None of the observed toilets provided toilet paper for patient use. Privacy for users was well-maintained in nearly all toilets, with concrete walls or partitions present in 100% of the facilities, and approximately 90% of them equipped with functional doors. Regarding facilities,

**Table 2. Descriptive statistics of selected hospitals (10 government and two non-government hospitals) in Dhaka, Bangladesh, July-December 2022.**

| Hospital type and codes | Year established | # of beds | # of patients[§] | # of caregivers and others[†] | # of staff | Total users[‡] | Total toilets | Overall: user-to-toilet ratio |
|---|---|---|---|---|---|---|---|---|
| **Govt. General hospitals** | | | | | | | | |
| H1 | 2001 | 250 | 433 | 2571 | 472 | 3476 | 132 | 26:1 |
| H2 | 2012 | 500 | 6746 | 12994 | 1159 | 20899 | 379 | 55:1 |
| **Govt. Medical college hospitals** | | | | | | | | |
| H3 | 1963 | 850 | 2612 | 6118 | 1548 | 10278 | 255 | 40:1 |
| H4 | 2009 | 500 | 4703 | 11504 | 1150 | 17357 | 430 | 40:1 |
| **Govt. specialized hospitals** | | | | | | | | |
| *Treat mainly infectious diseases* | | | | | | | | |
| H5 | 1972 | 300 | 503 | 1117 | 105 | 1725 | 43 | 40:1 |
| H6 | 2018 | 250 | 663 | 679 | 597 | 1939 | 194 | 10:1 |
| *Treat mainly chronic diseases* | | | | | | | | |
| H7 | 1978 | 414 | 2002 | 5454 | 995 | 8451 | 318 | 27:1 |
| H8 | 1982 | 300 | 2193 | 2965 | 715 | 5873 | 203 | 29:1 |
| H9 | 2012 | 450 | 1779 | 3343 | 1562 | 6684 | 183 | 37:1 |
| H10 | 2013 | 250 | 1011 | 2936 | 381 | 4328 | 177 | 24:1 |
| **Private medical college hospitals** | | | | | | | | |
| P1 | 1986 | 500 | 673 | 1089 | 378 | 2140 | 86 | 27:1 |
| P2 | 1992 | 500 | 178 | 314 | 334 | 826 | 59 | 14:1 |
| **Total/overall** | - | **5064** | **23496** | **51084** | **9396** | **83976** | **2459** | **34:1** |

*The total number of inpatient and outpatient present/visited on the day of data collection.

[†]Other users: security guard, canteen, and medical representatives.

[‡]Total number of users including staff, patients, caregivers, and other users use one toilet per day.

"H"-represented the codes for government hospitals.

"P"-represented the codes for private hospital.

we found that 84% of the toilet blocks had a minimum of one functional electric bulb, while only 8% had a functioning exhaust fan to maintain ventilation (Table 3).

We did not observe any signage indicating gender-segregated toilets for patient facilities. Only 3% (n = 97) of toilets had a bin for menstrual pad disposal and among them <1% (n = 8) bins had a functional lid. Overall, less than 1% (n = 10) of the toilets had partial facilities (only handles) for individuals with limited mobility. Due to lack of MHM facilities and the absence of separate toilets designated for individuals with limited mobility, none of the selected hospitals met basic sanitation service according to JMP (Table 3).

Our results revealed that both government and private hospitals had a significant number of toilets used by both male and female patients (i.e., mixed-gender toilet usage). Specifically, in government hospitals, 79% (n = 1839) of toilets were mixed-gender, followed by 11% (n = 244) dedicated to males and 10% (n = 231) exclusively for females. In private hospitals, 73% (n = 106) of toilets were mixed-gender, with 5% (n = 7) dedicated to males and 22% (n = 32) exclusively for females (Table 4).

## Toilet functionality

Table 4 provides a detailed analysis of toilet functionality and cleanliness based on different types of users. Overall, 68% (n = 1567) of toilets were observed functional, with 167% (n = 1216) of

**Table 3. Toilet observation status, overall toilet structure, functionality and water source for toilet use in 10 government hospitals and two non-government hospitals in Dhaka, Bangladesh, July-December 2022.**

| Variables | Number n (%) N = 2875 |
|---|---:|
| **Able to observe toilet,** yes | 2459 (86) |
| **Type of toilet pan** | |
| *Squatting pan flush toilet | 1083 (44) |
| †Sitting pan flush toilet | 1376 (56) |
| **Type of toilet floor** | |
| Ceramic tiles | 2409 (98) |
| Cemented floor | 30 (1) |
| Marble floor | 19 (1) |
| **Type of toilet flush** | |
| Cistern/tank flush (functional) | 1672 (68) |
| Cistern/tank flush (non-functional) | 443 (18) |
| Pour/bucket flush | 344 (14) |
| **Water seal,** Present | 2439 (99) |
| **Water use option for personal cleaning** | |
| Tap water to bucket/water pot | 1254 (51) |
| Hand shower | 1156 (47) |
| Only tap water but no bucket | 50 (2) |
| **Functional electric light present** | |
| Inside toilet block (Yes) | 2066 (84) |
| Inside toilet cubicle (Yes) | 1549 (63) |
| **Presence of function exhaust fan, yes** | 197 (8) |
| **Toilet maintain privacy, yes** | 2213 (90) |
| **Broken door (Yes)** | 246 (10) |
| **Toilets with solid disposal bin (MHM facility)** | 97 (3) |
| **Toilets for person with limited mobility (only handle)** | 10 (0.3) |

*A squat toilet (or squatting toilet) is a toilet used by squatting, rather than sitting. This means that the posture for defecation and for female urination is to place one foot on each side of the toilet drain or hole and to squat over it.
†A typical flush toilet is a ceramic bowl (pan) connected on the "up" side to a cistern (tank) that enables rapid filling with water, and on the "down" side to a drain pipe that removes the effluent.

toilets functional in inpatient services and 70% (n = 347) functional in outpatient services. A similar percentage of functionality was observed in government general hospitals (69%) and specialized hospitals (66%). Overall, high toilet functionality was observed [92% (n = 134)] in private hospitals, with 93% (n = 114) of toilets functional in inpatient services and 91% (n = 91) functional in outpatient services (Table 4). The highest level of functionality was observed in female toilets compared to mixed-gender toilets in both government and private hospitals (Fig 2). The lowest toilet functionality was observed in the diagnostics/pharmacy/blood bank areas of government hospitals (45%), followed by male staff (56%) and male patient toilets (59%) in specialized hospitals. In private hospitals, 100% of toilets were found to be functional in mixed-gender patient's ward toilets, and special care toilets. In government hospitals, only female staff toilets in general hospitals and medical college hospitals were 100% functional (Table 4).

## Toilet cleanliness

Overall, only 33% (n = 772) of toilets were observed to be clean, with 34% (n = 626) of toilets observed to be clean in inpatient services and 30% (n = 146) observed to be clean in outpatient

**Table 4. Descriptive statistics: functionality and cleanliness of toilets based on different types of users in 10 government hospitals and two non-government hospitals in Dhaka, Bangladesh, July-December 2022.**

| Type of hospital (# of toilets) | $^§$Functional toilets N/n (%) | | | $^{\|}$Clean toilets$^a$ | | |
|---|---|---|---|---|---|---|
| | Overall | Inpatients departments | Outpatients department | Overall, n (%) | Inpatients departments (%) | Outpatients department (%) |
| **All government hospitals (n = 2314 toilets)** | **1567 (68)** | **1216 (67)** | **346 (70)** | **772 (33)** | **626 (34)** | **146 (30)** |
| **Govt. general and medical college hospitals(n = 1196)** | **823 (69)** | **926/629 (68)** | **270/194 (72)** | **359 (30)** | **307 (33)** | **52 (19)** |
| Male (n = 141) | 92 (65) | 112/65 (58) | 29/27 (93) | 42 (30) | 32 (29) | 10 (34) |
| Female (n = 129) | 102 (79) | 112/88 (79) | 17/14 (82) | 45 (35) | 39 (35) | 6 (35) |
| Mixed-gender (n = 926) | 629 (68) | 702/476 (68) | 224/153 (68) | 272 (29) | 236 (34) | 36 (16) |
| $^†$**Staff (n = 96)** | **58 (60)** | **92/54 (59)** | **4/4 (100)** | **13 (14)** | **13 (14)** | **NA** |
| Male (n = 62) | 35 (56) | 61/34 (56) | 1/1 (100) | 6 (10) | 6 (10) | |
| Female (n = 6) | 6 (100) | 6/6 (100) | NA | 2 (33) | 2 (33) | |
| $^‡$Mixed-gender (n = 28) | 17 (61) | 25/14 (56) | 3/3 (100) | 5 (18) | 5 (20) | |
| $^*$**Patient's wards (n = 487)** | **339 (70)** | **479/332 (69)** | **8/7 (88)** | **164 (34)** | **162 (34)** | **2 (25)** |
| Male patients (n = 17) | 14 (82) | 17/14 (82) | NA | 10 (59) | 10 (59) | NA |
| Female patients (n = 50) | 39 (78) | 50/39 (78) | | 15 (30) | 15 (30) | |
| Mixed-gender (n = 420) | 286 (68) | 412/279 (68) | 8/7 (88) | 139 (33) | 137 (33) | 2 (25) |
| **Patient's service** | **256 (69)** | **347/235 (63)** | **26/21 (81)** | **133 (30)** | **132 (37)** | **1 (25)** |
| Cabins (n = 82) | 57 (70) | 82/57 (70) | NA | 52 (63) | 52 (63) | NA |
| Special care (n = 95) | 80 (84) | 90/76 (84) | 5/4 (88) | 16 (17) | 16 (18) | |
| Pediatric and gynae (n = 117) | 71 (61) | 114/71 (62) | 3/0 (0) | 43 (37) | 43 (38) | |
| Emergency (n = 10) | 9 (90) | NA | 10/9 (90) | 0 | NA | |
| Diagnostics/pharmacy/blood bank (n = 31) | 14 (45) | 27/10 (37) | 4/4 (100) | 9 (29) | 8 (30) | 1 (25) |
| Others (n = 38) | 25 (66) | 34/21 (62) | 4/4 (100) | 13 (34) | 13 (38) | NA |
| **Govt. specialized hospitals (n = 1118)** | **739 (66)** | **896/587 (66)** | **222/152 (68)** | **413 (37)** | **319 (36)** | **94 (42)** |
| Male (n = 103) | 74 (72) | 81/57 (70) | 22/17 (77) | 57 (55) | 46 (57) | 11 (50) |
| Female (n = 102) | 76 (75) | 79/61 (77) | 23/15 (65) | 66 (65) | 51 (65) | 15 (65) |
| Mixed-gender (n = 913) | 589 (65) | 736/469 (64) | 177/120 (68) | 290 (32) | 222 (30) | 68 (38) |
| **Staff and admin (112)** | **70 (63)** | **97/57 (59)** | **15/13 (87)** | **49 (44)** | **36 (37)** | **13 (87)** |
| Male (n = 35) | 21 (60) | 32/19 (59) | 3/2 (67) | 18 (51) | 16 (50) | 2 (67) |
| Female (n = 11) | 9 (82) | 8/7 (88) | 3/2 (67) | 9 (82) | 7 (88) | 2 (67) |
| Mixed-gender (n = 66) | 40 (61) | 57/31 (54) | 9/9 (100) | 22 (33) | 13 (23) | 9 (100) |
| **Patient's wards (n = 597)** | **359 (60)** | **561/338 (60)** | **36/21 (58)** | **174 (23)** | **168 (30)** | **8 (22)** |
| Male (n = 34) | 20 (59) | 31/20 (65) | 3/0 (0) | 16 (47) | 16 (52) | NA |
| Female (n = 52) | 34 (65) | 49/33 (67) | 3/1 (33) | 28 (54) | 27 (55) | 1 (33) |
| Mixed-gender (n = 511) | 305 (60) | 481/285 (59) | 30/20 (67) | 132 (26) | 125 (26) | 7 (23) |
| **Patient's service (n = 336)** | **264 (79)** | **230/186 (86)** | **106/78 (74)** | **168 (52)** | **111 (57)** | **57 (53)** |
| Cabins (n = 45) | 36 (80) | 45/36 (80) | NA | 30 (67) | 30 (67) | NA |
| Special care (n = 71) | 55 (77) | 71/55 (77) | NA | 29 (41) | 29 (41) | |
| Pediatric and gynae (n = 15) | 12 (80) | 5/5 (100) | 10/7 (70) | 8 (53) | 4 (80) | 4 (40) |
| Emergency (n = 19) | 17 (89) | 3/3 (100) | 16/14 (88) | 8 (42) | NA | 8 (50) |
| Diagnostics/pharmacy/blood bank (n = 161) | 126 (78) | 90/75 (83) | 71/51 (72) | 78 (48) | 39 (43) | 39 (55) |
| Other (n = 25) | 18 (72) | 16/12 (75) | 9/6 (67) | 15 (60) | 9 (56) | 6 (67) |
| **Private hospitals(n = 145)** | **134 (92)** | **123/114 (93)** | **22/20 (91)** | **81 (56)** | **70 (57)** | **11 (50)** |
| Male (n = 7) | 7 (100) | NA | 7/7 (100) | 2 (29) | 0 | 2 (29) |
| Female (n = 32) | **30 (94)** | 28/26 (93) | 4/4 (100) | 22 (69) | 18 (64) | 4 (100) |

(*Continued*)

**Table 4.** (Continued)

| Type of hospital (# of toilets) | §Functional toilets N/n (%) | | | ‖Clean toilets[a] | | |
|---|---|---|---|---|---|---|
| | Overall | Inpatients departments | Outpatients department | Overall, n (%) | Inpatients departments (%) | Outpatients department (%) |
| Mixed-gender (n = 106) | 97 (92) | 95/88 (93) | 11/9 (82) | 57 (54) | 52 (55) | 5 (45) |
| **Patient's wards (n = 73)** | **69 (95)** | **73/69 (95)** | NA | **49 (67)** | **49 (67)** | NA |
| Female (n = 14) | 14 (100) | 14/14 (100) | | 12 (86) | 12 (86) | |
| Mixed-gender (n = 59) | 55 (93) | 59/55 (93) | | 37 (63) | 37 (63) | |
| **Patients service (n = 53)** | **46 (87)** | **50/45 (93)** | **3/1 (33)** | **24 (51)** | **21 (48)** | NA |
| Special care (n = 11) | 11 (100) | 11/11 (100) | NA | 6 (55) | 6 (55) | |
| Pediatric and gynae (n = 25) | 21 (84) | 25/21 (84) | | 8 (32) | 8 (32) | |
| Diagnostics/pharmacy/blood bank (n = 8) | 6 (75) | 5/5 (100) | 3/1 (33) | 6 (75) | 3 (60) | |
| Others (n = 9) | 8 (89) | 9/8 (89) | NA | 4 (44) | 4 (44) | |

*Patients users: Patients including caregivers of hospitals reported by hospital authority during the day of data collection.

†Staff: toilet user's included doctors, nurse, and cleaning staff.

‡Mixed-gender: Toilet used by both males and females.

§Functional toilets: Toilet pan was not broken pan, functional door with lock, water available, not leakage of overhead sewage pipe, toilet pan was not overflowed or clogged.

‖Clean toilets: feces were not visible on any surface of the toilet, no sputum/cough, solid waste, no insects and rodent were present, floor was not submerged with water, no strong smell of feces.

[a]The denominator (N) for clean toilets are same as noted in the column under "Functional toilets" hence we did not repeat them.

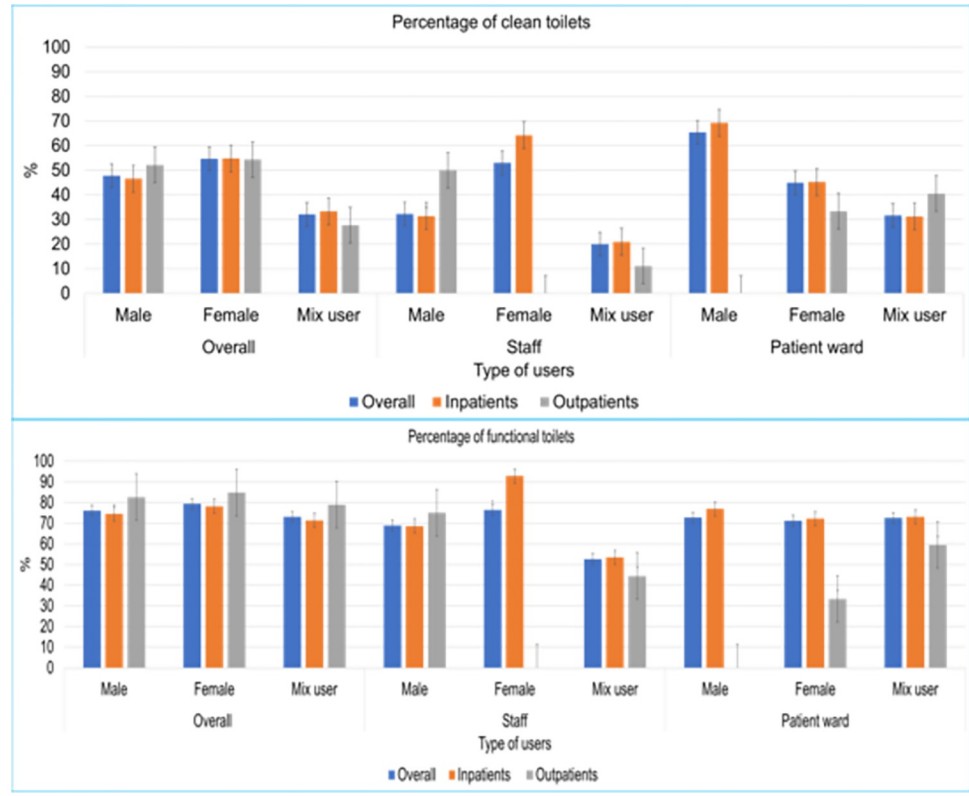

**Fig 2. Functionality and cleanliness status of the toilets in healthcare facilities in Dhaka city according to gender and patients service.**

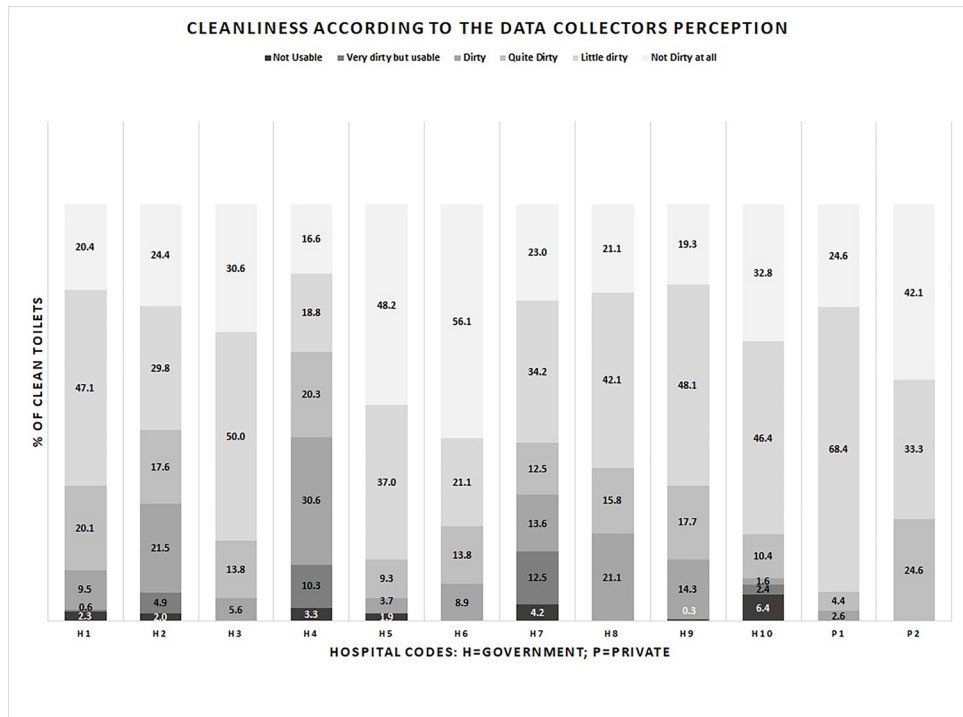

**Fig 3. Cleanliness of toilets according to data collector's perception during observation.**

services. Although a similar percentage of cleanliness was observed in government general hospitals (33%), toilet cleanliness was slightly high in specialized hospitals (37%). Overall, cleanliness was high [56% (n = 81)] in private hospitals, with 57% (n = 70) clean in inpatient services and 50% (n = 11) clean in outpatient services (Table 4). The highest level of cleanliness was observed in female toilets compared to mixed-gender toilets. Among staff toilets, mixed-gender toilets showed the lowest level of cleanliness in comparison to gender segregated (i.e., male and female) staff toilets (Fig 2). Private hospitals exhibited the highest cleanliness rates, with 100% (n = 9) of toilets observed as clean in outpatient shared toilets, followed by female staff outpatient facilities [88% (n = 7)] and female staff inpatient facilities [82% (n = 9)]. Conversely, in government hospitals, no clean toilets were found in emergency services, and only 16% of toilets were clean in mixed-gender patients' facilities. Cleanliness was also low in outpatients mixed-gender toilets (23%) and in the diagnostics/pharmacy/blood bank areas of government hospitals (25%) (Table 4). The perceived cleanliness of the toilets by the data collectors during toilet observation indicated that only 30% of toilets were reported as "not dirty at all," while more than 43% were reported as "little dirty," and the remaining 27% were described as "quite dirty" or "unusable" (Fig 3). Table 5 provides a detailed status of functionality and cleanliness for toilets in each hospital separately.

## User-to-toilet ratio

The overall user-to-toilet ratio was substantially higher in the outpatient service than in the inpatient service (Fig 4). Specifically, in the outpatient service of government hospitals, the user-to-toilet ratio was 214:1, and in private hospitals, the ratio was 94:1. Overall, the highest user-to-toilet ratio was 450:1 in the outpatients of a government specialized hospital (H5; 300 beds), while the lowest ratio was 74:1 in the inpatients of a government general hospital (H6;

**Table 5. User-to-toilet ratio, functionality, and cleanliness of toilets in different types of HCFs in Dhaka, Bangladesh, July-December 2022.**

| Type of hospital (# of toilets) | User-to-toilet ratio | | | | | | [§]Functional toilets n (%) | | [‖]Clean toilets n (%) | |
|---|---|---|---|---|---|---|---|---|---|---|
| | Inpatient departments | | | Outpatient departments | | | Inpatient departments | Outpatient departments | Inpatient departments | Outpatient departments |
| | Patients[*]: toilet | Staff[†]: toilet | Shared[‡]: toilet | Patients: toilet | Staff: toilet | Shared: toilet | | | | |
| **Government hospitals (n = 2314)** | **17:1** | **12:1** | **19:1** | **214:1** | **22:1** | **183:1** | **1216 (67)** | **346 (70)** | **626 (34)** | **146 (30)** |
| **General hospitals (n = 511)** | **6:1** | **11:1** | **24:1** | **251:1** | **13:1** | **131:1** | **236 (64)** | **118 (81)** | **123 (34)** | **30 (20)** |
| H1(n = 132) | 4:1 | 36:1 | 51:1 | 0 | 11:1 | 42:1 | 61 (68) | 27 (64) | 23 (26) | 0 |
| H2 (n = 379) | 7:1 | 5:1 | 16:1 | 251:1 | 14:1 | 195:1 | 175 (63) | 91 (88) | 100 (36) | 30 (29) |
| **Medical College hospitals (n = 685)** | **20:1** | **10:1** | **15:1** | **172:1** | **68:1** | **264:1** | **393 (70)** | **76 (61)** | **184 (33)** | **22 (18)** |
| H3 (n = 255) | 36:1 | 14:1 | 23:1 | 175:1 | 14:1 | 197:1 | 154 (72) | 28 (68) | 56 (26) | 5 (12) |
| H4 (n = 430) | 10:1 | 7:1 | 11:1 | 172:1 | 92:1 | 34:13 | 239 (69) | 48 (57) | 128 (37) | 17 (20) |
| **Specialized hospitals (n = 1118)** | **18:1** | **14:1** | **19:1** | **202:1** | **8:1** | **158:1** | **587 (66)** | **152 (68)** | **319 (36)** | **94 (42)** |
| *Treat mainly infectious diseases* | | | | | | | | | | |
| H5 (n = 43) | 12:1 | 8:1 | 0 | 450:1 | 3:1 | 0 | 27 (69) | 4 (100) | 10 (26) | 1 (25) |
| H6 (n = 194) | 10:1 | 7:1 | 15:1 | 74:1 | 4:1 | 0 | 106 (65) | 16 (50) | 76 (47) | 11 (34) |
| *Treat mainly chronic diseases* | | | | | | | | | | |
| H7 (n = 318) | 25:1 | 8:1 | 21:1 | 375:1 | 13:1 | 939:1 | 191 (64) | 8 (42) | 60 (20) | 7 (37) |
| H8 (n = 203) | 23:1 | 70:1 | 21:1 | 114:1 | 7:1 | 13:1 | 56 (56) | 63 (61) | 21 (21) | 48 (47) |
| H9 (n = 183) | 14:1 | 13:1 | 41:1 | 264:1 | 22:1 | 48:1 | 113 (80) | 39 (95) | 74 (52) | 19 (46) |
| H10 (n = 177) | 31:1 | 5:1 | 14:1 | 101:1 | 3:1 | 368:1 | 94 (61) | 22 (96) | 78 (51) | 8 (35) |
| **Private hospitals (n = 145)** | **9:1** | **13:1** | **18:1** | **94:1** | **32:1** | **14:1** | **114 (93)** | **20 (91)** | **70 (57)** | **11 (50)** |
| P1 (n = 86) | 11:1 | 12:1 | 13:1 | 172:1 | 37:1 | 30:1 | 72 (99) | 11 (85) | 54 (74) | 7 (54) |
| P2 (n = 59) | 3:1 | 16:1 | 21:1 | 0 | 5:1 | 4:1 | 42 (84) | 9 (100) | 16 (32) | 4 (44) |
| **Overall total (n = 2459)** | **17:1** | **12:1** | **19:1** | **205:1** | **22:1** | **176:1** | **1330 (68)** | **366 (71)** | **696 (36)** | **157 (31)** |

[*]Patients users: Patients including caregivers of hospitals reported by hospital authority during the day of data collection.

[†]Staff: toilet user's included doctors, nurse, and cleaning staff.

[‡]Shared facility (staff and patients): Toilet used by both patients, caregivers and staff.

[§]Functional toilets: Toilet pan was not broken pan, functional door with lock, water available, not leakage of overhead sewage pipe, toilet pan was not overflowed or clogged.

[‖]Clean toilets: feces were not visible on any surface of the toilet, no sputum/cough, solid waste, no insects and rodent were present, floor was not submerged with water, no strong smell of feces.

250 beds). The overall user-to-toilet ratio for staff was lower in both government and private hospitals compared to the user-to-toilet ratio for patients (Fig 4). In the inpatient service of government hospitals, the user-to-toilet ratio was 17:1, and in private hospitals, the ratio was 9:1. The highest user-to-toilet ratio was 36:1 in the inpatients of a government medical college hospital (H3; 850 beds), while the lowest ratio was 4:1 in the inpatients of a government general hospital (H1; 250 beds). The highest user-to-toilet ratio for staff was 70:1 in the inpatients of a government specialized hospital (H8; 300 beds), while the lowest ratio was 5:1 in the

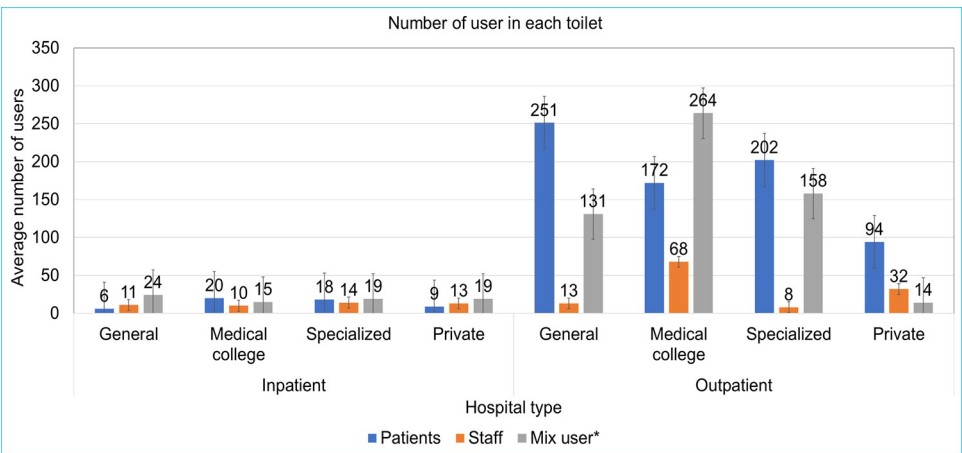

**Fig 4. Average number of users per toilet in different patient service and in different types of hospitals in Dhaka city.**

inpatients of both a government general hospital (H2; 500 beds) and a government specialized hospital (H10; 250 beds) (Table 5). User-to-toilet ratio further increased when we considered only functional toilets in the analysis (S1 Table in S1 File).

### Factors affecting functionality and cleanliness of toilets

Our multivariate regression model suggested that private hospitals exhibited significantly higher rates of functional toilets (OR = 3.73, p<0.001) and clean toilets (OR = 8.49, p<0.001) than government hospitals. Similarly, hospitals constructed after 2000 (i.e., newly constructed buildings) demonstrated significantly higher functionality and cleanliness compared to those built before 2000 (i.e., relatively older buildings). Our data also suggested that toilets used by > 30 users (high number of users) significantly lower functionality (OR = 0.71, p<0.005) and cleanliness rates (OR = 0.44, p<0.001) than those with fewer than 30 users. Staff-used toilets displayed significantly higher levels of functionality (OR = 1.66, p<0.001) and cleanliness (OR = 2.28, p<0.001) than patient-used toilets. While toilets for females had higher functionality (OR = 1.36, p = 0.132), no difference in cleanliness was observed (OR = 1.43, p = 0.057) compared to male toilets. Additionally, no significant difference in functionality and cleanliness were observed between inpatient and outpatient services (Table 6).

### Discussion

This hospital-based study presents a comprehensive analysis of sanitation facilities in large HCFs in Dhaka, Bangladesh. It covers availability, accessibility, functionality and cleanliness of sanitation facilities, and user-to-toilet ratios in government and private hospitals within urban low-resource settings, as well as considering gender and inclusion aspects. The study also employed a novel methodology and data collection tools for precise toilet observation, accurately estimating user-to-toilet ratios among patient/caregiver, staff, and mixed-gender users separately in complex HCFs. Moreover, this study included both government and private hospitals, covering various categories such as general, medical college, and specialized hospitals, thus ensuring the comprehensive representation of hospital types in Dhaka city. To our knowledge, there is a lack of studies estimating toilet conditions and appropriate user-to-toilet ratios in large urban hospitals in LMICs, which is important for Bangladesh due to higher patient loads in Dhaka compared to smaller cities [17,32–34]. While previous studies in Bangladesh

**Table 6. Toilet functionality and cleanliness of selected hospitals in 10 government hospitals and two non-government hospitals in Dhaka, Bangladesh, July-December 2022.**

| Hospital type (# of toilet) | Functional toilets, n | % | (OR, p-value)* | Clean toilets, n | % | (OR, p-value)* |
|---|---|---|---|---|---|---|
| Government (N = 2314) | 1562 | 77.50 | | 772 | 33.36 | |
| Private (N = 144) | 134 | 92.41 | (3.73, p<0.001) | 81 | 55.85 | (8.49, p<0.001) |
| **Year constructed** | | | | | | |
| Before 2000 (N = 714) | 513 | 71.83 | | 247 | 34.56 | |
| After 2000 (N = 1750) | 1311 | 75.00 | (1.46, p<0.001) | 540 | 36.57 | (1.24, p<0.05) |
| **Type of users** | | | | | | |
| Patient's toilet (N = 871) | 606 | 69.57 | | 240 | 27.55 | |
| Staff toilets (N = 1054) | 837 | 79.40 | (1.66, p<0.001) | 582 | 55.21 | (2.28, p<0.001) |
| Shared facility (N = 539) | 381 | 70.70 | (1.06, p = 0.641) | 119 | 22.08 | (0.93, p = 0.620) |
| **Patient's service** | | | | | | |
| Inpatients (N = 1949) | 1,413 | 72.50 | | 721 | 37.00 | |
| Outpatients (N = 515) | 411 | 79.81 | (1.13, p = 0.271) | 166 | 32.25 | (1.04, p = 0.734) |
| **Gender** | | | | | | |
| Male (N = 222) | 169 | 76.12 | | 106 | 47.75 | |
| Female (N = 276) | 219 | 79.34 | (1.36, p = 0.132) | 151 | 54.71 | (1.43, p = 0.057) |
| Mix-gender (N = 1966) | 1436 | 74.57 | (1.05, p = 0.765) | 630 | 32.00 | (0.89, p = 0.476) |
| **# of users per toilet** | | | | | | |
| ≤30 (N = 1869) | 1432 | 66.61 | | 801 | 42.86 | |
| >30 users (434) | 302 | 69.60 | (0.71, p<0.005) | 54 | 12.44 | (0.44, p<0.001) |

*p value is generated by multiple logistic regression adjusting relevant co-variates.

and elsewhere in Africa and Asia assessed overall HCF WASH [34–37], they rarely analyzed user-to-toilet ratio alongside functionality and cleanliness. A study assessed WASH in HCFs in Kampala, Uganda but lacked data on hospital size and daily patient load [36]. A nationwide Bangladesh study comprehensively analyzed HCF WASH, yet only reported toilet type, location, and overall cleanliness [34]. The paper categorized toilets using the 2019 WHO-UNICEF JMP classifications for improved toilets [5]. Our current study provides comprehensive and detailed exploration of sanitation facilities in large HCFs within an urban low-resource setting. The findings and insights gained from this study provide important new evidence for policy-makers, healthcare administrators and other researchers working in this field.

We consistently observed a user-to-toilet ratio in excess of 10 in outpatient services and in excess of 100 for outpatient services across the selected hospitals in Dhaka. At its most extreme, a user-to-toilet ratio of 939:1 was observed in an outpatient toilet within a government hospital. Similarly, both patient and mixed-gender toilets consistently displayed increased ratio compared to staff toilets, whether in inpatient or outpatient services. Currently, there is a lack of guidance at local level on the appropriate user-to-toilet ratio in HCFs, including specific guidance for staff and patients. Furthermore, there is no information available regarding the implications of these high user-to-toilet ratios for public health, infection prevention and control (IPC), and patient experience [17]. According to WHO-UNICEF Water and Sanitation for Health Facility Improvement Tool (WASH FIT) 2022 global technical guidelines, achieving an appropriate user-to-toilet ratio involves having a dedicated toilet for women and men separately in each toilet block [18]. Additional toilets should be provided at a ratio of one per 20–25 patients/caregivers, with an extra toilet added for every 50 additional patients/caregivers [22]. For outpatient and caregiver services, maintaining a 2:1 ratio of female to male toilets is recommended [20]. According to the Bangladeshi national WASH standard and

implementation guidelines from 2021, hospitals are recommended to maintain a ratio of 6:1 (1 toilet for every 6 inpatient beds) to ensure sufficient sanitation facilities [38,39]. The recent WASH FIT report recommended having at least two or more improved toilets for outpatients, plus an additional toilet per 20 users/inpatients [18].

Our present toilet assessment indicates that while the average user-to-toilet ratio across all HCFs aligned with the WHO WASH FIT (user-to-toilet ratio in inpatients service: 20:1) [18] guidelines for staff and patients/caregiver's toilets in the inpatient service, it did not meet the required standard ratio for outpatient toilets and mixed-gender toilets in the inpatient service. Our data revealed that in the outpatients service of government hospitals, overall user-to-toilet ratio was 214:1, and in private hospitals, the ratio was 94:1. There was no guideline on the appropriate user-to-toilet ratio at local level for HCFs [38,39]. Hospitals need to reassess their toilet infrastructure to match the actual patient loads, particularly in outpatient areas. To tackle the issue of high user-to-toilet ratio, adopting a multi-sectoral approach in healthcare facility design is crucial. As far back as 2008, other authors have called for collaborative efforts between healthcare, architecture, and urban planning sectors can result in well-designed building layouts that ensure toilets are distributed appropriately according to patient loads, while also incorporating gender, and disability-friendly facilities [40]. This comprehensive approach will ultimately lead to enhanced sanitation services and improved user experiences within healthcare facilities.

The functionality rates observed in this study were unacceptably low. Our findings indicated that approximately one-third of the toilets (31%) were non-functional, and two-thirds (65%) were unclean. The HCFs in Dhaka are already grappling with an excessively high number of toilet users. The prevalence of broken and unclean toilets exacerbates this situation, rendering it challenging and impractical to ensure safe sanitation services for both patients and staff. While the Joint Monitoring Programme (JMP) provides a clear definition for toilet functionality, which includes factors like a toilet not being broken, the toilet pit not being blocked, and availability of water for flush/pour-flush toilets [1], the definition of a clean toilet for HCFs is not adequately outlined. The WASH FIT report also categorised the toilet quality using colours (green, yellow and red), but lacks an adequate definition of toilet cleanliness. Although the WHO's guide on environmental cleaning and IPC in HCFs in LMICs provides guidelines on how to clean toilets, it does not explicitly define clean and unclean toilets [13]. Notably, Netherlands Development Organisation (SNV)'s recent sanitation impact indicators and assessment tools for Bangladeshi HCFs includes toilet cleanliness as a success indicator to define hygienic sanitation facilities in such settings [33]. It is important to emphasize that toilet hygiene and cleanliness are closely intertwined with overall environmental or surface cleaning in hospitals and consider toilet hygiene as a part of overall IPC in HCFs [13,36].

Several studies have concluded that unhygienic toilets in HCFs can contribute to the transmission of various diseases [7–12]. Our findings revealed that approximately 70% of government-affiliated toilets were found to be unclean, with around 27% of these facilities being notably dirty and unusable. Recent research has shown that public hospitals in Dhaka often face overcrowding issues, surpassing their patient care facility capacities (such as the number of available beds, waiting areas, and space for attendants) [17]. This strain negatively impacts on sanitation facilities, including insufficient hand hygiene infrastructure and inadequate infection control measures. Moreover, reports suggest that major hospitals in Dhaka lack adequate cleaning staff to maintain the prescribed IPC protocols [1]. Taken together, these congested conditions coupled with limited WASH amenities may heighten the likelihood of transmitting infectious diseases, like enteric and respiratory diseases both from patients to their caregivers and vice versa, as well as among asymptomatic caregivers and other patients [7,32,41].

Our findings also suggested that several factors were associated with lower levels of toilet cleanliness such as high user-to-toilet ratio (in cases where toilets are used by more than 30 users), the age of the toilet facilities (with older buildings showing lower cleanliness), and the usage of mixed-gender toilets as opposed to gender-segregated ones. Furthermore, our results revealed that government-run toilets were significantly less clean in comparison to those in private hospitals. Our results also revealed that toilets designated for patients were notably less clean than those designated for staff. A multisectoral collaborative approach is necessary to ensure appropriate user-to-toilet ratio for both staff and patients, modify infrastructure to accommodate more patient toilets, especially in outpatient services, and develop and adopt separate cleaning protocols for sanitation facilities, particularly in large hospitals, incorporate sanitation and cleanliness standards [17,33].

The study found that gendered sanitation needs were not addressed. A significant number of mixed-gender toilets was evident across all hospitals, and there was an absence of signage designating gender-segregated toilets (male/female) in any of the hospital toilet blocks. This observation highlights that despite separate designs for male and female toilets, there was consistent overlapping of male and female patients in all hospitals. This situation compromises the privacy of female patients, caregivers, and hospital staff. As noted in other studies, it is crucial to have separate toilet blocks for women due to the need for more space, privacy, and time during urination and defecation, especially considering menstruation [24]. Ensuring privacy and safety is also crucial to prevent any increase in the risk of violence against women and girls or any feelings of vulnerability among users. User perception also plays a significant role; if a facility is perceived as unsafe, it can discourage usage and drive individuals towards potentially less hygienic alternatives [42].

Gendered needs as regards MHM were also not met. We found that only 3% toilets had a trash bin for menstrual pad and solid disposal and among them <1% bin had functional lid. The lack of MHM facilities in HCFs not only affects the physical health and hygiene of females but also has broader implications for gender equality, patient care, and the overall well-being of individuals and working in healthcare settings [43–45]. Furthermore, sewerage systems in hospital are designed to carry water and fecal matter; discarding solid waste such as menstrual pads and cloth in toilets can block sewage pipes and led to non-functional toilets [30]. This also poses challenges for sewage treatment plants and can have significant cost and human resource.

From an inclusion perspective, needs for persons with limited mobility were also not met. Our data indicates that no hospital had a toilet facility for people with limited mobility. To achieve basic sanitation in HCFs, each section/toilet block (male and female) should ideally have at least one accessible toilet cubicle for individuals with disabilities that meets national or international [1] standards. Hospitals admit a wide range of people with limited mobility every day [46]. However, globally, there is a lack of data concerning the number, types, and severity of disabled individuals in HCFs, particularly in LMICs [46]. People with physical disabilities face a higher risk of falls and accidents when there are no toilets designed specifically for them. The absence of disability-friendly toilets in each toilet block can make it difficult for individuals with disabilities to use the facilities independently [47]. This lack of accessible toilets can result in challenges, including dependency, stress, embarrassment, discomfort, and difficulties in self-care. The absence of suitable toilet options may lead individuals with disabilities to avoid healthcare facilities, causing delayed care and negative health outcomes [47]. Given that none of the hospitals in Dhaka city addressed these critical indicators, none of the selected hospitals met the criteria for basic sanitary facilities. Ensuring toilets with MHM facilities for women and dedicated toilets for individuals with limited mobility are two essential components for achieving the basic sanitation standards outlined by the JMP in HCFs [48].

Although our study possessed significant strengths, it also has four key limitations. Firstly, we did not include private hospitals systematically, primarily focusing on government health-care facilities. While this approach ensured representation of government hospitals in Dhaka city, it limited insights into the sanitation situation in private hospitals. The inclusion of only two hospitals did not adequately represent the diversity within private hospitals in Dhaka city. Future research should include a more comprehensive and representative sample of private hospitals to offer a holistic view. Secondly, we couldn't observe 14% of toilets either because they were locked or due to restricted access permissions. The results of this study might vary if we had included the unobserved toilets. Third, data were collected at a specific time point between June and December 2022, spanning from the post-monsoon to the winter season. We were unable to account for potential variations in patient flow during various seasons or dis-ease outbreaks. Previous studies conducted in HCFs in Dhaka demonstrated substantial increases in patient flow during epidemics as well as seasonal variation [49,50]. To gain a more comprehensive understanding, it is crucial to monitor patient flows consistently throughout the year and evaluate the seasonal fluctuations in toilet usage, as this could have implications for toilet cleanliness and functionality. Finally, the study was conducted after the COVID-19 pandemic, a period during which several hospitals were shifting from treating COVID-19 cases (i.e., COVID dedicated hospitals) to focusing on general medical care (i.e., all general hospitals). This transition may have influenced patient and caregivers flow and, as a result, could have affected toilet usage patterns and user-to-toilet ratios.

## Conclusions and recommendations

This research found sanitation facilities in hospitals in Dhaka to be inadequate from a number of perspectives. The HCFs in Dhaka city are grappling with a significant influx of patients and caregivers utilizing the toilets, which has led to a strain on the existing sanitation systems. The toilets designated for patients in outpatient services are facing an exceptionally high user load, warranting immediate attention and intervention to alleviate this situation. The prevalence of non-functional and unclean toilets further exacerbates the sanitation challenges within these hospitals. Additionally, the lack of gender-segregated toilets and menstrual hygiene manage-ment (MHM) facilities and lack of toilets for individuals with limited mobility in all 12 hospi-tals underscores the critical gaps in sanitation infrastructure. Without addressing these challenges, the goal of achieving basic and safe sanitation facilities for patients of hospitals and HCFs in dense urban settings such as Dhaka, and particularly for women and individuals with limited mobility by 2030 appears to be unattainable. The government should allocate resources for the appropriate design of sanitation facilities in HCFs that meet acceptable user ratios rec-ommended by the recent WHO WASH-FIT guidelines for both in-and outpatient services as well as for healthcare professionals, ensuring an adequate number of sanitation workers and providing proper training on toilet hygiene and IPC to improve the sanitation facilities in HCFs. Multisectoral approaches and adequately funded plans are urgently needed to improve the availability, functionality, and cleanliness of toilets in hospitals in Bangladesh and LMICs more broadly. Doing so will be critical to protecting public health and accelerating efforts towards Sustainable Development Goal 6 (SDG6).

## Supporting information

**S1 File. S1 Fig and S1 Table.**
(DOCX)

## Acknowledgments

We would like to convey our gratitude to the Directorate General of Health Services (DGHS) under Bangladesh's Ministry of Health and Family Welfare (MoHFW) for providing approval and necessary support to conduct the research. We acknowledged the Director, Hospital and Clinic Sections, DGHS, for his strong leadership and continuous support in collecting samples and conducting the study in selected COVID-19 and non-CVID19 hospitals. We are also thankful to the Directors, doctors, and support staff of all selected hospitals for their permission and support during sample and data collection, interviewing, and effective implementation at the facility. We are also indebted to icddr,b fieldworkers workers who contributed immensely to conducting field activities to make this study successful. icddr,b is also grateful to the governments of Bangladesh and Canada, for providing core/unrestricted support.

## Author Contributions

**Conceptualization:** Nuhu Amin, Tim Foster, Supriya Sarkar, Aninda Rahman, Mahbubur Rahman, Juliet Willetts.

**Data curation:** Nuhu Amin, Md. Imam Hossain.

**Formal analysis:** Nuhu Amin, Md. Imam Hossain.

**Funding acquisition:** Nuhu Amin, Juliet Willetts.

**Investigation:** Md Rezaul Hasan.

**Methodology:** Nuhu Amin, Tim Foster, Md. Imam Hossain, Md Rezaul Hasan, Shaikh Daud Adnan, Juliet Willetts.

**Project administration:** Nuhu Amin, Md. Imam Hossain, Md Rezaul Hasan, Supriya Sarkar, Aninda Rahman, Shaikh Daud Adnan, Mahbubur Rahman.

**Resources:** Nuhu Amin, Shaikh Daud Adnan, Mahbubur Rahman, Juliet Willetts.

**Software:** Nuhu Amin.

**Supervision:** Tim Foster, Shaikh Daud Adnan, Juliet Willetts.

**Validation:** Nuhu Amin, Tim Foster, Supriya Sarkar, Aninda Rahman, Juliet Willetts.

**Visualization:** Nuhu Amin.

**Writing – original draft:** Nuhu Amin.

**Writing – review & editing:** Tim Foster, Md. Imam Hossain, Md Rezaul Hasan, Supriya Sarkar, Aninda Rahman, Shaikh Daud Adnan, Mahbubur Rahman, Juliet Willetts.

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
