## [Decision Letter · Decision Letter 0]

2 Jan 2024

PONE-D-23-39815Inadequate Sanitation in Healthcare Facilities: A Comprehensive Evaluation of Toilets in Major Hospitals in Dhaka, BangladeshPLOS ONE

Dear Dr. Amin,

Thank you for submitting your manuscript to PLOS ONE. After careful consideration, we feel that it has merit but does not fully meet PLOS ONE’s publication criteria as it currently stands. Therefore, we invite you to submit a revised version of the manuscript that addresses the points raised during the review process.

Although the four reviewers are broadly supportive of the paper, there is a lot of work to do on definitions and terminology, particularly from reviewer 3.   It will take some time to go through carefully but I think worthwhile.

We look forward to receiving your revised manuscript.

Kind regards,

Alison Parker

Academic Editor

PLOS ONE

Journal Requirements:

2. Thank you for stating the following financial disclosure: "This project has been funded by the Bill & Melinda Gates foundation (BMGF) [grant number INV-045360]. "

3. We note that your Data Availability Statement is currently as follows: "All relevant data are within the paper and its Supporting Information files."

4. Please upload a copy of Supporting Information Figure/Table/etc. Fig S1 which you refer to in your text on page 34.

Reviewers' comments:

Reviewer's Responses to Questions

**Comments to the Author**

1. Is the manuscript technically sound, and do the data support the conclusions?

Reviewer #1: Yes

Reviewer #2: Partly

Reviewer #3: Partly

Reviewer #4: Partly

2. Has the statistical analysis been performed appropriately and rigorously? 

Reviewer #1: Yes

Reviewer #2: No

Reviewer #3: I Don't Know

Reviewer #4: Yes

3. Have the authors made all data underlying the findings in their manuscript fully available?

Reviewer #1: No

Reviewer #2: Yes

Reviewer #3: Yes

Reviewer #4: Yes

4. Is the manuscript presented in an intelligible fashion and written in standard English?

Reviewer #1: Yes

Reviewer #2: Yes

Reviewer #3: No

Reviewer #4: Yes

5. Review Comments to the Author

Reviewer #1: This paper has provided a detailed case study of health care facility sanitation access in Dhaka, and where there are gaps the authors provide suggestions on what needs to be done to improve the situation on the ground. The data is well presented and clearly written. The references are up-to-date and look to be complete. The study limitations is well presented. In Figure 1, add that Private Hospitals were purposively selection. Please ensure Figure 2 and 3 were developed with a statistical package such as R, and not Excel. Figure 4 has some fonts that are unclear. Figure S1 is really wonderful. There is a need to include as supplementary material the raw data underlying this study.

Reviewer #2: The is need for pointing out the gap within the introduction to provide a strong reason why the study was necessary. Point out the novelty in the study. provide study hypothesis and study limitations. Provide inclusion and exclusion criteria. Include Indepth statistical analysis.

Reviewer #3: I applaud the initiative on undertaking this study and believe it is a useful piece of work.

However, I feel the paper is undermined by unclear text, too much verbiage, ambiguous statements, and a lack of rigour in definitions.

One area in which there is ambiguity is the way the JMP definitions have been used. I note that the JMP uses the following terms: "The term usable here refers to toilets or latrines that are accessible to patients and staff (doors are unlocked or a key is available at all times), functional (the toilet is not broken, the toilet hole is not blocked, and water is available for flush/pour-flush toilets), and private (there are closable doors that lock from the inside and no large gaps in the structure)". However, the authors use the term functional without explaining that is is a sub-category of "usable", and also do not spell out the definition at the point it is first used (which is very early in the paper). When the authors state that the recent global report shows that a number of HCF toilets are "non-functionable" and unusable" (pg 2), this would be a good place to state the definitions.

The third para on page 3 is very muddled, as it starts out talking about women and people with disabilities, but then mixes in children, and ends with the needs of menstruating women but no further mention of people with limited mobility. I would suggest having a stand-along para just on the needs of women, and then mention the needs of people with limited mobility separately (and also the needs of children if you want to include them). A small point here is that you have mentioned needs during menstruation, but not the similar needs of women during postpartum bleeding, which would be particularly relevant in HCFs.

in the Study Design paper on page 4, once again terms are introduced with no definition. The text states that "functionality" and "cleanliness" were assessed but it is not explained that "cleanliness" is not part of the JMP definition. Cleanliness should be defined here.

I found the text on "cabins" and "toilet blocks" confusing. What was the smallest unit of toilet used? a toilet "seat" (whether raised commode for sitting or a squatting plate)? can a toilet have multiple seats or cabins? this is not clear and as the term "toilet" is used exensively throughout I suggest this be made clearer. Is a "toilet block" multiple cabins or seats? What is the significance of a toilet block? A small point, but I also found some of the terminology unclear - for instance, I am not sure what a "medicine ward" is. Perhaps this is a term specific to Bangladeshi or Asian hospitals but it might not be recognized by every reader.

Again, the authors might want to note where they have differed from, or added to, the JMP definitions. For instance, the JMP does not include the presence of a water seal in its definition.

The cleanliness scale includes the term "not usable" which is confusing as the JMP has a very specific meaning for "usable" which does not encompass cleanliness.

I found the term "toilet user" confusing - was this the number of people who actually used a toilet on the day of the data collection, or potential users, that is people who are at the clinic or hospital who might reasonably be expected to want to use the toilet while they are there? Again, specifying this would help.

It was hard to appreciate the statistical nature of the analysis given there was was so little clarity on definitions. For instance, the JMP definition of functionality is not spelled out until page 22.

The text mentions that 10% of toilets could not be observed as they were locked - but this is part of the JMP definition of accessible, so is worthy of more comment.

The number of hand washing basins with soap is mentioned, but not not those that had water but no soap (which is another rung on the hygiene ladder)

By water for "personal cleaning" do the authors mean anal cleansing? good to say so if so, this is a more universally recognized term. I assume this water is not for handwashing.

Table 2 could be improved by putting the percentages first, and the n numbers in brackets -this is a more usual presentation. I was not impressed with the footnotes with definitions for squat and flush toilets from Wikipedia! these definitions could be stated earlier (without reference to Wikipedia)

The text on gender segregated toilets was very confusing - are toilets defined as gender segregated if there is signage? the text seems to imply this but it would be better to state it explicitly. And do the authors mean that toilets designated for women were nonetheless used by men as well or that some toilets were designed to be used by both men and women - which of these is what "mixed gender toilet usage" means?

On page 11 the text states that none of the hospitals met "basic sanitary facilities" according to the JMP, but this is not the terminology the JMP uses - providing a "basic sanitation service" is the JMP term.

I got a bit lost in the large amount of text that basically re-stated the numbers in the tables. I felt much of it was unnecessary and repetitive.

One of my biggest criticisms of the paper is the mistake in reporting whether a ratio was high or low. The authors consistently referred to a ratio with a small numerator and a large denominator as "high". That is a LOW ratio. The point the authors want to make is that the number of toilets relative to the number of users is low, but this is presented as a "high" ratio throughout - this is incorrect and this mistake undermines the credibility of the paper. Likewise, on page 21, line 505, it is reported that functionality rates were unacceptably high - when in fact they are LOW.

The text that restates the figures in the tables rapidly becomes boring...I would have much preferred commentary on the figures that were anomalies, or particularly noteworthy. On page 18 there is a statement that "interestingly" toilets used by more than 30 users showed lower functionality and cleanliness - but no explanation of why this is interesting (in fact, it seems predictable).

Some of the more useful definitions and standards appear in the discussion, for instance WaterAid's technical guidelines, whereas I think they would have been more usefully presented earlier in the paper. And on page 22 the JMP definition finally appears, but alongside what seems to be a criticism of JMP and WHO for not having a clear definition of a "clean" toilet in HCFs - this is a bit buried but an important point if well presented and backed up.

I note that the recommendations state that resources should be allocated for appropriate design, and that "acceptable" user ratios should be met, but there is no clear recommendation as to what those ratios should be - this would make the recommendations much stronger.

In short, this is an interesting piece of research, but could be much better presented and explained. I urge the authors to establish clear definitions upfront, and state the standards that should be met, and then explain clearly why the research showed that toilets in hospitals and healthcare facilities are not meeting these standards.

Some small things:

-there seems to be a typo on pg 3, line 98 1427/230 is not 16%.

-The term "patient's service" on page 7 line 237 is not clear.

Reviewer #4: Introduction: There are some duplicate references in bibliography and going back and forth between years of data from same source (JMP) that should be revisited. Also, important to clearly define terminology in introduction - what is meant by basic sanitation and how the study defines functionality.

Methods: Further details are needed on dates of data collection, selection criteria, how some data was collected with observation checklist (# of patients, caregivers, staff using toilets, which staff used which toilets, etc.).

Results: It is not clear how some of the results were obtained, as it is not clearly defined in the methods. For example, # of patients, caregivers and staff. Was this collected through observation or through interviews? For some questions such as sex segregation, it is also not clear how this was obtained. Was one person in the facility interviewed or multiple persons?

In general, some editing of text is suggested in comments to improve readability and avoid confusion.

6. PLOS authors have the option to publish the peer review history of their article (what does this mean?). If published, this will include your full peer review and any attached files.

Reviewer #1: No

Reviewer #2: No

Reviewer #3: No

Reviewer #4: No

---

## [Author Response · Author response to Decision Letter 0]

7 Mar 2024

Response to the reviewers’ comments: 

Dear Reviewers,

Thank you for providing constructive feedback on the manuscript titled “Inadequate Sanitation in Healthcare Facilities: A Comprehensive Evaluation of Toilets in Major Hospitals in Dhaka, Bangladesh.” Your suggestions have significantly improved the clarity of the manuscript. We have carefully reviewed the comments and addressed them accordingly. Please find our responses in both track change format and clean version to the specific queries below:

Review Comments to the Author

Reviewer #1: 

This paper has provided a detailed case study of health care facility sanitation access in Dhaka, and where there are gaps the authors provide suggestions on what needs to be done to improve the situation on the ground. The data is well-presented and clearly written. The references are up-to-date and look to be complete. The study limitations is well presented. In Figure 1, add that Private Hospitals were purposively selection. Please ensure Figure 2 and 3 were developed with a statistical package such as R, and not Excel. Figure 4 has some fonts that are unclear. Figure S1 is really wonderful. There is a need to include as supplementary material the raw data underlying this study.

Response: Dear Reviewer, thank you for your constructive feedback and suggestions for further improvements to the manuscript. Please see our specific responses bellow: 

1. In Figure 1: add that Private Hospitals were purposively selected.

Response: We have added “purposively” as suggested in Figure 1. 

2. Figure 2: Please ensure Figure 2 and 3 were developed with a statistical package such as R, and not Excel.

Response: We agree it would have been great if we could develop Figure 1 & 2 with a statistical package such as R. Due to the complexity and shapes of our data, we cannot develop Figure 2 & 3 with such a package. Moreover, currently, we do not have any members who can develop the graphs using R. However, we have tried our best to improve the visualization. Since other reviewers agreed with the current graphics, we did not make substantial changes.

3. Figure 4: Figure 4 has some fonts that are unclear.

Response: We agree some fronts are unclear in Figure 4 (currently Figure 3) due to overlapping of data. We have corrected the inconsistency in the revised version. In the current version Figure 4 is labelled as “Figure 3” due to rearrange the texts. 

4. Raw data in the supplementary materials: There is a need to include as supplementary material the raw data underlying this study.

Response: We agree with you, and we already included as supplementary material the raw data underlying this study. 

Reviewer #2: 

The is need for pointing out the gap within the introduction to provide a strong reason why the study was necessary. Point out the novelty in the study. provide study hypothesis and study limitations. Provide inclusion and exclusion criteria. Include In-depth statistical analysis.

Response: Dear Reviewer, thank you for your positive and concise feedback on the manuscript. The following are the responses related to your comments:

1. Include gaps within the introduction to provide a strong reason why the study was necessary.

Response: We appreciate your suggestion. We already pointed out the research gaps and the justification of the study. 

“There is a notable lack of global data on toilet functionality and hygiene, especially in LMICs such as Bangladesh. In Bangladesh, recent sanitation evaluations have focused on sub-district level hospitals, neglecting larger HCFs in major cities. To address these knowledge gaps in HCFs, we conducted a cross-sectional study to comprehensively explore the availability, functionality, cleanliness, user-to-toilet ratios, and adequacy of sanitation facilities in 10 government and two private hospitals in Dhaka, Bangladesh.” 

“The study focused on addressing existing research gaps and providing insights to enhance sanitation services and infection prevention in a high-density urban context in Bangladesh, with implications for other similar LMIC settings.” Please see on page 6, lines 158-172.

2. Point out the novelty of the study.

Response: From our recent literature review, we found that there is a lack of data on toilet conditions and appropriate user-to-toilet ratios in large urban hospitals in LMICs. Very few studies in HCFs have highlighted the overall sanitation status in LMICs but they did not quantify the actual knowledge gaps such as user-to-toilet ratio, percentages of toilets for women and persons with limited mobility, etc. Furthermore, we have used unique method to estimate actual user-to-toilet for staff, patients, male, female separately. Previous studies used proxy indicators to estimate the number. Please see the novelty of the study mentioned in the method and discussion section below: 

Introduction: 

There is a notable lack of global data on toilet functionality and hygiene, especially in LMICs such as Bangladesh. In Bangladesh, recent sanitation evaluations have focused on sub-district level hospitals, neglecting larger HCFs in major cities. No study has estimated the user-to-toilet ratio for different user groups (i.e., staff, patients, and caregivers) in major healthcare facilities (HCFs) in Bangladesh. Furthermore, there is a lack of data on sanitation facilities for persons with limited mobility and menstrual hygiene management (MHM) facilities for women in HCFs at local level. Please see on page 6, lines 162-168.

Materials and Methods: See “Defining of latrine blocks in each hospital floor” section on page 10-11 and “Estimation of the toilet user” section on pages 14-15. 

Discussion: 

“This hospital-based study presents a comprehensive analysis of sanitation facilities in large HCFs in Dhaka, Bangladesh. It covers availability of sanitation facility, functionality, cleanliness, and user-to-toilet ratios in government and private hospitals within urban low-resource settings, as well as considering gender and inclusion aspects. The study also employed a novel methodology and data collection tools for precise toilet observation, accurately estimating user-to-toilet ratios among patient/caregiver, staff, and mixed-gender users separately in complex HCFs. Moreover, this study included both government and private hospitals, covering various categories such as general, medical college, and specialized hospitals, thus ensuring the comprehensive representation of hospital types in Dhaka city.” please see the texts on page 33 and lines 655-664.

3. Provide study hypothesis and study limitations.

Response: The study was cross-sectional study aiming to assess the hospital sanitation status in HCFs, so we believe it does not need a study hypothesis. Regarding the study limitation, we already provided detailed study limitations in the final paragraph of the discussion section. If you have any specific queries related to the limitation, we are happy to address them. 

4. Provide inclusion and exclusion criteria.

Response: Thank you for your suggestion. We used robust methods to select the tertiary care HCFs in Dhaka city. We included hospitals from a wide geographic region and representing almost all types of tertiary care hospitals in Dhaka city. We have added the inclusion and exclusion criteria in the method section as follows: 

“For this study, we only enrolled major tertiary care hospitals in Dhaka city. We included at least one hospital from each category (general, specialized, or medical college and hospitals) and of different sizes (small, medium, or large based on the number of beds). Additionally, we considered varied geographic locations within the Dhaka South City Corporation (DSCC) and Dhaka North City Corporation (DNCC) areas. A total of ten government hospitals were selected. This included two general hospitals, two medical college hospitals, and six specialized hospitals. For comparison purposes, we purposively selected two private medical college hospitals (Fig 1) to establish a reference group alongside the government hospitals. We did not include any primary care hospital, and community clinics in this study.” Please see on page 7, lines 193-207.

5. Include In-depth statistical analysis.

Response: To our knowledge, we already performed several statistical tests to validate our findings. First, we performed Shapiro-Wilk, Shapiro-Francia, and Skewness and Kurtosis tests to determine whether the data was normally distributed or not. We also performed a two-sample Z test of proportions to determine whether the toilet functionality and cleanliness differ significantly between inpatient and outpatient services. In addition, we used univariate and multivariate logistic regression models to evaluate the effects of multiple factors on toilet functionality and cleanliness. Hosmer–Lemeshow goodness-of-fit test was used to assess how well the model fit for the analyzed data. Variance Inflation Factor (VIF) was performed to test the multicollinearity among the explanatory variables used in the model. Moreover, the multivariate model was adjusted for co-variates such as user type, gender, user number per toilet to overcome the confounding effects. The mixed-effects logistic regression model was also adjusted with robust standard error to control the clustering effect within hospitals. However, if you have any specific suggestion beyond these analyses, we are happy to address them.

Reviewer #3: 

I applaud the initiative on undertaking this study and believe it is a useful piece of work.

However, I feel the paper is undermined by unclear text, too much verbiage, ambiguous statements, and a lack of rigour in definitions.

Response: Dear reviewer, Thank you for your thoughtful review and valuable suggestions to enhance the clarity of the manuscript. We have carefully addressed your comments and incorporated the suggested revisions into the updated version of the manuscript.

One area in which there is ambiguity is the way the JMP definitions have been used. I note that the JMP uses the following terms: "The term usable here refers to toilets or latrines that are accessible to patients and staff (doors are unlocked or a key is available at all times), functional (the toilet is not broken, the toilet hole is not blocked, and water is available for flush/pour-flush toilets), and private (there are closable doors that lock from the inside and no large gaps in the structure)". However, the authors use the term functional without explaining that is a sub-category of "usable", and also do not spell out the definition at the point it is first used (which is very early in the paper). When the authors state that the recent global report shows that a number of HCF toilets are "non-functionable" and unusable" (pg 2), this would be a good place to state the definitions.

Response: Thank you for your insightful feedback regarding the usage of JMP definitions in our manuscript. We acknowledge the ambiguity in the way the terms "usable" and "functional" were presented. In response to your suggestion, we have included a separate section outlining the operational definitions, and within the introduction and method sections, we explicitly provide the definitions of these terms as per the JMP criteria. 

Note that for the data collection in our study hospitals, JMP definition was not sufficient. For example, functionality was defined as “the toilet is not broken, the toilet hole is not blocked, and water is available for flush/pour-flush toilets”. In this definition the terminology “toilet is not broken” contain many sub-variable including broken door, broken pan, or broken overhead sewage pipes (which was common problem in Dhaka), etc,. For this reason, we have used extensions of JMP definitions where applicable. We also added a number of new operational definitions to describe different sanitation related terminologies in this manuscript. We believe these revisions enhanced the clarity and precision of our terminology. Please see table 1 on pages 8-9. 

“usable toilets” in HCFs refer to toilets or latrines that are accessible to patients and staff (doors are unlocked or a key is available at all times), and “functional toilet” refers to a toilet that is not broken or blocked, and water is available for flush/pour-flush toilets [5]

The third para on page 3 is very muddled, as it starts out talking about women and people with disabilities, but then mixes in children, and ends with the needs of menstruating women but no further mention of people with limited mobility. I would suggest having a stand-along para just on the needs of women, and then mention the needs of people with limited mobility separately (and also the needs of children if you want to include them). A small point here is that you have mentioned needs during menstruation, but not the similar needs of women during postpartum bleeding, which would be particularly relevant in HCFs.

Response: We have restructured the entire section to include two standalone paragraphs—one addressing the needs of women and another focusing on the needs of people with limited mobility. This adjustment aims to improve clarity and organization in presenting different aspects. We did not collect relevant data on the needs of children, so we have removed the term "children" from the paragraph. In response to your suggestion, we have added a mention of postpartum bleeding alongside menstrual hygiene management (MHM) to ensure a more comprehensive coverage of relevant topics. Please see the changes on pages 4, lines 123-140.

In the Study Design paper on page 4, once again terms are introduced with no definition. The text states that "functionality" and "cleanliness" were assessed but it is not explained that "cleanliness" is not part of the JMP definition. Cleanliness should be defined here.

Response: This suggestion is addressed along with previous responses. Please see the changes on pages 8-9 and in the method section. 

I found the text on "cabins" and "toilet blocks" confusing. What was the smallest unit of toilet used? a toilet "seat" (whether raised commode for sitting or a squatting plate)? can a toilet have multiple seats or cabins? this is not clear and as the term "toilet" is used exensively throughout I suggest this be made clearer. Is a "toilet block" multiple cabins or seats? What is the significance of a toilet block? A small point, but I also found some of the terminology unclear - for instance, I am not sure what a "medicine ward" is. Perhaps this is a term specific to Bangladeshi or Asian hospitals but it might not be recognized by every reader.

Response: We apology for this confusion. A cabin is a room where only a patient is treated and there is one dedicated toilet for that patient. Toilet blocks we defined based on different situations where only one toilet can be a toilet block again multiple toilets can be a toilet block. We added and now revised based on your comments how we defined toilet blocks in the methods section giving the heading “Defining toilet blocks in each hospital floor” as well as defining the medicine ward and cabin. Please see the changes on page 10-11. Also see definitions on page 8-9. 

The significance of a toilet block was to observe all toilets of an HCF in a systematic way so that none of the toilets missed out from the observation.

Again, the authors might want to note where they have differed from, or added to, the JMP definitions. For instance, the JMP does not include the presence of a water seal in its definition.

Response: You are correct. The JMP definitions did not entirely align with the definitions used in this study to explore the current sanitation situation. For instance, JMP defines functionality as the toilet not being broken, the toilet hole not being blocked, and water being available for flush/pour-flush toilets. It also defines privacy as having closable doors that lock from the inside and no large gaps in the structure. In our study, we found instances where the toilet pans were not physically broken but were blocked with solid particles, rendering them non-functional. We also observed cases where the toilet pan itself was functional, but due to blockages in sewage pipes, feces overflowed we defined the toilet as “non-functional”. Furthermore, some toilets

---

## [Decision Letter · Decision Letter 1]

8 Apr 2024

PONE-D-23-39815R1Inadequate Sanitation in Healthcare Facilities: A Comprehensive Evaluation of Toilets in Major Hospitals in Dhaka, BangladeshPLOS ONE

Dear Dr. Amin,

Thank you for submitting your manuscript to PLOS ONE. After careful consideration, we feel that it has merit but does not fully meet PLOS ONE’s publication criteria as it currently stands. Therefore, we invite you to submit a revised version of the manuscript that addresses the points raised during the review process.

 **Reviewer 3 had provided some further minor comments to be addressed.**

We look forward to receiving your revised manuscript.

Kind regards,

Alison Parker

Academic Editor

PLOS ONE

Journal Requirements:

Reviewers' comments:

Reviewer's Responses to Questions

**Comments to the Author**

1. If the authors have adequately addressed your comments raised in a previous round of review and you feel that this manuscript is now acceptable for publication, you may indicate that here to bypass the “Comments to the Author” section, enter your conflict of interest statement in the “Confidential to Editor” section, and submit your "Accept" recommendation.

Reviewer #3: (No Response)

2. Is the manuscript technically sound, and do the data support the conclusions?

Reviewer #3: Yes

3. Has the statistical analysis been performed appropriately and rigorously? 

Reviewer #3: I Don't Know

4. Have the authors made all data underlying the findings in their manuscript fully available?

Reviewer #3: Yes

5. Is the manuscript presented in an intelligible fashion and written in standard English?

Reviewer #3: Yes

6. Review Comments to the Author

**Reviewer #3**: Congratulations to the authors for addressing all the comments. I find the paper much improved, and am happy to recommend it be accepted.

I have a few small suggests for final edits:

-the JMP is mentioned in line 68 with the word "also", but the JMP has not been introduced in the text at this point in the paper

-I would be careful about suggesting that the JMP "recommends" anything (line 99) - the JMP has definitions and indicators, but is itself not a normative agency

-line 101 mentions WHO-JMP but this is not correct - I think you mean WHO-UNICEF JMP

-the text on lines 103-107 is in the wrong place - should be in a separate section on study limitations or with other text describing the study methodology

-on line 178 I think the text is supposed to say "we adopted....the JMP definitions..." the word definitions is missing

-WASH FIT is mentioned in line 180 but has not been described - it is introduced much later in the paper -not until line

-line 364 mentions that <1% of bins had a functional lid - this is such a small percentage that perhaps it makes sense to mention the exact number (I assume in single digits!)

-I not that for some of the tables the edit I suggested, having the percentage first and the n number in parentheses, has been adopted, but not in Table 4 or Table 5. Up to the editor to direct you, but I would be inclined to fix this inconsistency

-not clear why you used a 2015 JMP report for classifications - this is the end of the MDG era - perhaps you need to make this clear - that these are not the same classifications that are used currently by JMP

-line 521 says there is a lack of "data" on the appropriate user to toilet ratio but of course data don't tell you what is appropriate or not...pehraps you mean there is a lack of "guidance" on the appropriate ratio?

-line 537 says the "user to toilet ratio aligned" but do you mean the average ratio across all HCF studied? if so, specify...otherwise this is vague

-line 541 -data "suggest"? shouldn't this be "showed" or "revealed"?

-line 591 is awkward - "toilet cleanliness exhibited"?

-I don't think the Uganda example mentioned in line 595 is relevant - not HCF toilets (so cooperation among households not relevant)

-line 624 seems to suggest problems at sewage treatment plants have an impact on dignity - the reference to dignity does not belong here - maybe in line 619?

7. PLOS authors have the option to publish the peer review history of their article (what does this mean?). If published, this will include your full peer review and any attached files.

Reviewer #3: No

---

## [Author Response · Author response to Decision Letter 1]

13 Apr 2024

Dear Reviewer,

Thank you for providing additional feedback on the manuscript titled “Inadequate Sanitation in Healthcare Facilities: A Comprehensive Evaluation of Toilets in Major Hospitals in Dhaka, Bangladesh.” Your suggestions have significantly improved the clarity of the manuscript. We have carefully reviewed the comments and addressed them accordingly. Please find our responses in both track change format and clean version to the specific queries below:

Review Comments to the Author

Reviewer #3: Congratulations to the authors for addressing all the comments. I find the paper much improved, and am happy to recommend it be accepted.

Response: Much appreciate your time and efforts providing valuable suggestions to improve the manuscript. 

I have a few small suggests for final edits:

-the JMP is mentioned in line 68 with the word "also", but the JMP has not been introduced in the text at this point in the paper. 

Response: Removed “also” from the sentence. 

-I would be careful about suggesting that the JMP "recommends" anything (line 99) - the JMP has definitions and indicators, but is itself not a normative agency

Response: Thinks for the suggestion. I have removed the JMP recommendation and added WaterAid users-to-toilet information. 

-line 101 mentions WHO-JMP but this is not correct - I think you mean WHO-UNICEF JMP

Response: Thanks again for your careful review. We have corrected accordingly. 

-the text on lines 103-107 is in the wrong place - should be in a separate section on study limitations or with other text describing the study methodology

Response: We have removed the section. 

-on line 178 I think the text is supposed to say "we adopted....the JMP definitions..." the word definitions is missing

Response: Corrected accordingly. 

-WASH FIT is mentioned in line 180 but has not been described - it is introduced much later in the paper -not until line

Response: We have elaborate/define the WASH-FIT. 

-line 364 mentions that <1% of bins had a functional lid - this is such a small percentage that perhaps it makes sense to mention the exact number (I assume in single digits!)

Response: added (n=8)

-I not that for some of the tables the edit I suggested, having the percentage first and the n number in parentheses, has been adopted, but not in Table 4 or Table 5. Up to the editor to direct you, but I would be inclined to fix this inconsistency

Response: I would like to keep the number first and then the percentage [n (%)]. Both of the style is acceptable globally including PlosOne policy. 

-not clear why you used a 2015 JMP report for classifications - this is the end of the MDG era 

Response: We have used 2019 classification from WASH data. We corrected the year and put citation.

- perhaps you need to make this clear - that these are not the same classifications that are used currently by JMP

Response: I think 2015 was a typo. We used latest classification by JMP WASH data (washdata.org/monitoring/sanitation)

-line 521 says there is a lack of "data" on the appropriate user to toilet ratio but of course data don't tell you what is appropriate or not...pehraps you mean there is a lack of "guidance" on the appropriate ratio?

Response: Corrected the wording and used “guidance”

-line 537 says the "user to toilet ratio aligned" but do you mean the average ratio across all HCF studied? if so, specify...otherwise this is vague

Response: Thanks for your suggestion. You are right this was the average of all HCFs. Corrected accordingly, 

“Our present toilet assessment indicates that while the average user-to-toilet ratio across all HCFs aligned with the WHO WASH FIT”

-line 541 -data "suggest"? shouldn't this be "showed" or "revealed"?

Response: Corrected and used “revealed”.

-line 591 is awkward - "toilet cleanliness exhibited"?

Response: Corrected accordingly. 

“Our results also revealed that toilets designated for patients were notably less clean than those designated for staff.”

-I don't think the Uganda example mentioned in line 595 is relevant - not HCF toilets (so cooperation among households not relevant)

Response: We agree with you and removed the sentence accordingly. 

-line 624 seems to suggest problems at sewage treatment plants have an impact on dignity - the reference to dignity does not belong here - maybe in line 619?

Response: Thanks for the suggestion. We have revised accordingly.

---

## [Editor Report · Decision Letter 2]

17 Apr 2024

Inadequate Sanitation in Healthcare Facilities: A Comprehensive Evaluation of Toilets in Major Hospitals in Dhaka, Bangladesh

PONE-D-23-39815R2

Dear Dr. Amin,

We’re pleased to inform you that your manuscript has been judged scientifically suitable for publication and will be formally accepted for publication once it meets all outstanding technical requirements.

Kind regards,

Alison Parker

Academic Editor

PLOS ONE
---

## [Editor Report · Acceptance letter]

26 Apr 2024

PONE-D-23-39815R2 

PLOS ONE

Dear Dr. Amin, 

I'm pleased to inform you that your manuscript has been deemed suitable for publication in PLOS ONE. Congratulations! Your manuscript is now being handed over to our production team.

Kind regards, 

on behalf of

Dr. Alison Parker 

Academic Editor

PLOS ONE